# CBP-HSF2 structural and functional interplay in Rubinstein-Taybi neurodevelopmental disorder

Patients carrying autosomal dominant mutations in the histone/lysine acetyl transferases CBP or EP300 develop a neurodevelopmental disorder: Rubinstein-Taybi syndrome (RSTS). The biological pathways underlying these neurodevelopmental defects remain elusive. Here, we unravel the contribution of a stress-responsive pathway to RSTS. We characterize the structural and functional interaction between CBP/EP300 and heat-shock factor 2 (HSF2), a tuner of brain cortical development and major player in prenatal stress responses in the neocortex: CBP/EP300 acetylates HSF2, leading to the stabilization of the HSF2 protein. Consequently, RSTS patient-derived primary cells show decreased levels of HSF2 and HSF2-dependent alteration in their repertoire of molecular chaperones and stress response. Moreover, we unravel a CBP/EP300-HSF2-N-cadherin cascade that is also active in neurodevelopmental contexts, and show that its deregulation disturbs neuroepithelial integrity in 2D and 3D organoid models of cerebral development, generated from RSTS patient-derived iPSC cells, providing a molecular reading key for this complex pathology.

Many genes associated with risk for neurodevelopmental disorders (NDDs) encode chromatin-modifying enzymes and other epigenetic regulators (e.g., the monogenic Rett syndrome and the heterogenous spectrum of autism disorders). Accordingly, a growing body of evidence suggests that the development and/or maintenance of cognitive abilities are dependent on chromatin and epigenetic regulation[1,2]. Illustrating this, Rubinstein-Taybi Syndrome (RSTS) is a rare monogenic, autosomal dominant NDD, characterized by brain abnormalities, intellectual disabilities at diverse degrees of severity, and multiple congenital malformations[3–5]. Mutations in the histone/lysine acetyl transferases (HATs/KATs) *CREBBP* (CREB-binding protein or CBP; KAT3A; RSTS1, OMIM #180849) and *EP300* genes (E1A-binding protein p300; KAT3B; RSTS2, OMIM #613684) represent 60% and 8–10% of clinically diagnosed RSTS cases, respectively[3]. These enzymes have key roles in neurodevelopment[4,6] and do not only acetylate histones, but also a number of non-histone proteins, including many transcription factors[7,8]. The presence of one mutated allele of either CBP or EP300 is sufficient to provoke this very disabling disease, which is suggestive of

haploinsufficiency or dominant-negative effects[4,9]. Although recent studies using RSTS patient-derived cells have underlined neuronal morphological and excitability defects[10], it has been difficult to decipher the molecular consequences of CBP or EP300 deficiency in the RSTS brain[4,9]. Indeed, despite the fact that global decrease in histone acetylation has been detected in patients and mouse models of RSTS[4,6], inhibiting the enzymes responsible for deacetylation (histone/lysine deacetylases; HDACs), whilst improving behavioral defects in mouse models and motor skills in patients, does not ameliorate patient cognitive functions[11]. Moreover, no clear genotype-phenotype correlation has been established, and for a given mutation or deletion, unexplained phenotypic variability is frequent. Altogether, this suggests that unknown biological pathways might be implicated in shaping the RSTS neurodevelopmental deficits, in addition to the direct effects of CBP or EP300 mutations on the chromatin landscape. Such pathways might constitute important therapeutic targets.

We and others have unraveled a crucial role of a stress-responsive pathway, active in normal neurodevelopment, and have shown that, in

✉e-mail: aurelie.dethonel@univ-paris-diderot.fr; valerie.mezger@univ-paris-diderot.fr

models of NDDs of environmental origin including prenatal alcohol or heavy metal exposure, its dysregulation leads to neurodevelopmental defects[12–16]. Heat-shock transcription factors (HSFs) were originally identified and characterized due to their stress responsiveness and ability to recognize a consensus DNA-binding site, the heat-shock element (HSE). HSFs are activated by various stressors (such as heat, oxidative stress, alcohol) and govern the expression of molecular chaperones (including Heat Shock Proteins, HSPs), a process called the heat-shock response, which contributes to maintenance and recovery of protein homeostasis, i.e., proteostasis[17,18]. In addition to their classical functions as stress-sensors and proteostasis guardians, and as integrators of stimuli shaping developmental, metabolic, and lifespan pathways, HSFs perform a broad spectrum of roles under both physiological and pathological conditions[18,19]. HSFs regulate the expression of a wide repertoire of target genes beyond the *HSPs*. As a consequence, abnormal HSF protein levels and/or activity are pivotal to susceptibility to metabolism, inflammation, cancer, and neurological disorders, via dysregulation of their diverse target genes[18]. The multifaceted roles of HSFs are achieved by their multimodular structure and oligomerization, post-translational modifications (PTMs), as well as a diversity of partner networks in a stress- and context-dependent manner. In particular, as regulators of gene expression, HSFs interact with and recruit chromatin regulators to their target genes, thus acting as mediators of epigenetic processes and sculptors of the epigenetic landscape[8,17,18,20]. The versatile functions of HSFs have been mostly studied with two members of the mammalian HSF family, HSF1 and HSF2[18]. HSF2 is a short-lived protein and its levels are tightly regulated in a proteasome-dependent manner[21]. It is highly abundant and active during prenatal brain development, where it controls the number of radial glia cells and radial neuronal migration in the neocortex by regulating a family of genes involved in microtubule dynamics[13–15,19,22]. Our recent study revealed that HSF2 also regulates a large number of genes belonging to the *Cadherin* superfamily, which plays pleiotropic roles in neurodevelopment[23–27]. While no specific function has been attributed to HSF1 during the prenatal development, in physiological conditions, it is involved in spinogenesis and neurogenesis during the postnatal development of the murine hippocampus[28]. In line with the roles of HSF1 and HSF2 in neurodevelopment, dysregulation of the HSF pathway has been observed in models of NDDs, i.e., affective and depressive-like behaviors and autism[16,28] and in neurodegenerative disorders[18,29].

Here, we show that the mutated CBP or EP300 in cells derived from RSTS patients compromise the integrity and functionality of the HSF pathway, thereby impacting the stability of the short-lived HSF2 protein. Using complementary biophysical, biochemical, cellular, and in silico structural approaches, we demonstrate that CBP/EP300 mediates the acetylation of HSF2 on specific lysine residues, through interaction between a defined (KIX) domain of anchorage in CBP with the HSF2 oligomerization domain, which promotes the stabilization of HSF2. As a consequence of decreased HSF2 protein stability, we show that RSTS patient-derived cells are impaired in their stress responsiveness and display reduced ability to express genes that are critical for neurodevelopment and regulated by HSF2, including the *N-cadherin* gene. We provide evidence that pharmacological or genetical rescue of HSF2 levels in RSTS primary cells restores both the stress response and neurodevelopmental gene expression. We find that the disruption of the CBP/EP300-HSF2-N-cadherin pathway is recapitulated in RSTS patient-derived neuroprogenitor cells (iNPCs) and human cerebral organoids (hCOs), which display proliferation abnormalities resembling those caused by impaired cell–cell adhesion, in particular in the N-cadherin pathway. The neurodevelopmental and stress-responsive facets of the HSF2 pathway thus provide a conceptual framework for understanding the molecular basis of the complex RSTS pathology.

## Results

### CBP/EP300 interacts with and acetylates HSF2 in neural models

We first analyzed the expression patterns of HSF2 and CBP/EP300 in hCOs[30] at day 56 of differentiation (D56). Using immunofluorescent labeling, we observed that both HSF2 and CBP/EP300 proteins were expressed in nuclei of the hCO proliferative and neuronal layers (PL, stained by SOX2 or PAX6; and NL, stained by TBR1 or beta III-TUBULIN) (Fig. 1a, b). Similarly, in the mouse, HSF2 was expressed in the germinal and neuronal areas of the neocortex, i.e., the ventricular zone (VZ) and cortical layer (CL) (Supplementary Fig. 1a). HSF2, CBP, and EP300 proteins, as well as their corresponding gene transcripts were present from D20 to D60 of hCO differentiation (Fig. 1c, Supplementary Fig. 1b), and at all stages of the developing mouse cortex from E11 to E17 (Supplementary Fig. 1c). Note that, in line with our findings in the mouse cortex[14,15], HSF1 was also expressed in D20-D60 hCOs (Fig. 1c, Supplementary Fig. 1c). We also found that HSF2 and CBP/EP300 colocalized in nuclei of proliferative or neuronal layers of hCOs. Moreover, both CBP and p300 were co-immunoprecipitated with HSF2 in the developing mouse cortex, and HSF2 was present in an acetylated form not only in the developing mouse cortex, but also in D40 hCOs (Fig. 1d–f, Supplementary Fig. 1d). Similar results were obtained in the human neural cell line SHSY-5Y (Supplementary Fig. 1e). Altogether, these results indicate that HSF2 and CBP/EP300 co-exist in the nuclei of both proliferative and neuronal cells, and that these proteins can interact in the neurodevelopmental context, which is correlated to the presence of HSF2 in an acetylated form in mouse prenatal cortices and hCOs.

### CBP/EP300 promotes HSF2 acetylation on key lysine residues

Since exogenous tagged HSF2 was acetylated in HEK 293 cells (Fig. 1f), we examined whether HSF2 was a substrate of CBP/EP300, using human HEK 293 cells co-expressing CBP-HA or EP300-HA and GFP- or Myc-tagged HSF2. We found that the immunoprecipitated exogenous HSF2 protein was acetylated by EP300 or CBP (Fig. 2a), but not by a dominant-negative CBP, unable to catalyze acetylation (Fig. 2b). To identify the acetylated lysine residues in HSF2, HSF2 was immunoprecipitated from HEK 293 cells co-expressing Flag-HSF2 with EP300-HA and lysine acetylation was analyzed by mass spectrometry (MS). Among the 36 lysines of HSF2, we identified 8 acetylated residues: K82 (located in the DNA-binding domain), K128, K135, K197 (all three located within the oligomerization HR-A/B domain), K209, K210, K395, and K401 (Fig. 2c, Supplementary Fig. 2a, Supplementary Table 1 and Supplementary data 1). Single-point mutations (K82, K128, K135, and K197), or the mutation of the doublet K209/K210 to arginine (R), which prevents acetylation, did not abolish global HSF2 acetylation (Supplementary Fig. 2b). This suggests that, in line with our MS data, the acetylation of HSF2 occurs on more than a single lysine residue. Accordingly, we defined several key acetylated lysine residues whose mutations to either arginine (R) or glutamine (Q) dramatically reduced HSF2 acetylation: K82, K128, K135, and K197 (Fig. 2d, Supplementary Fig. 2c). To dissect the requirement of CBP in the acetylation of HSF2, we used an in vitro acetylation assay coupled with high-performance liquid chromatography (HPLC). We found that a synthetic HSF2 peptide, containing either K135 or K197 residue was readily acetylated, in an acetyl-CoA-dependent manner, by the purified recombinant full-catalytic domain of CBP (Full-HAT; Fig. 3a), whereas a peptide containing K82 was not acetylated (Figs. 2e, f, and 3a, Supplementary Fig. 2d–f). Note that it was not possible to analyze the HSF2 K128 peptide due to its insolubility (Manufacturer's information). Taken together, our data suggest that HSF2 is acetylated by CBP/EP300 at three main lysine residues (K128, K135, and K197), which are located within the oligomerization HR-A/B domain.

Prompted by the finding that a catalytically active CBP is necessary for HSF2 acetylation, we examined whether HSF2 could bind to

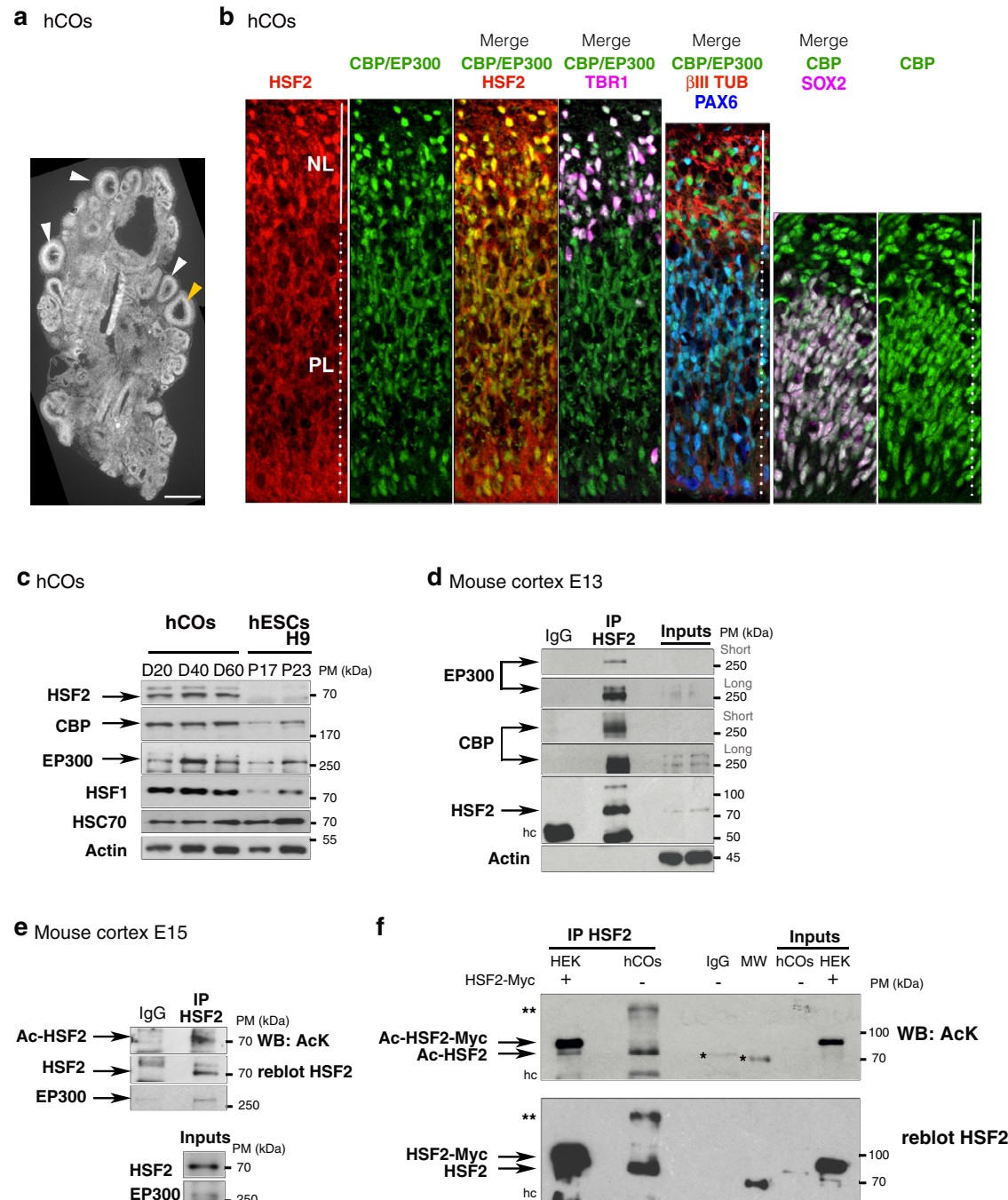

**Fig. 1 | HSF2 is expressed, acetylated, and interacts with EP300/CBP in neurodevelopmental contexts.** Representative images or immunoblots. **a** Confocal microscopy images of 56 days (D56) hCOs derived from H9 human embryonic stem cells (hESCs) and stained with DAPI. Image reconstruction of a complete section showing cortical-like structures (white arrowheads). The yellow arrowhead points to magnified areas shown in **b** (panels 1–4). Scale bar: 200 µm. **b** Immunofluorescence of D56 hCOs showing the co-expression of HSF2, CBP/P300 in PAX6 or SOX2 neuroprogenitor cells and in TBR1 or class III β-TUBULIN (βIII tub) neurons (*n* = 3). Top, basal side; Bottom, apical side. Dotted lines, PL proliferative layer, solid lines NL neuronal layer. Each panel is 70 µm wide. **c** Immunoblots from D20, D40, and D60 hCOs and H9 hESCs at passages 17 and 23, showing CBP/EP300, HSFs and HSC70 expression (*n* = 3). HSC70, a heat-shock cognate protein that is not induced by stress,

serves as a loading control. Actin is used as a comparison. **d** Immunoblots of immunoprecipitated HSF2 (IP HSF2), showing co-immunoprecipitation of EP300 and CBP (*n* = 3) in the mouse cortex at embryonic day 13 (E13). hc IgG heavy chain. Inputs, total proteins in input samples. Short and long exposure times. **e** Immunoblots of immunoprecipitated HSF2 showing endogenous acetylated HSF2 (Ac-HSF2) in the mouse cortex at E15 (*n* = 2). Co-immunoprecipitation of EP300 is used as a positive control. AcK acetyl-lysine. **f** Immunoblots of immunoprecipitated HSF2 from HEK 293 cells overexpressing a myc-tagged HSF2 or D40 hCOs hESCs showing acetylation of HSF2 (*n* = 3). HEK 293 cells are positive controls that contain both endogenous and exogenous acetylated HSF2. *indicates non specific, **indicates high molecular weight form of HSF2. MW molecular weight, hc IgG heavy chain. Source data are provided as a Source Data file.

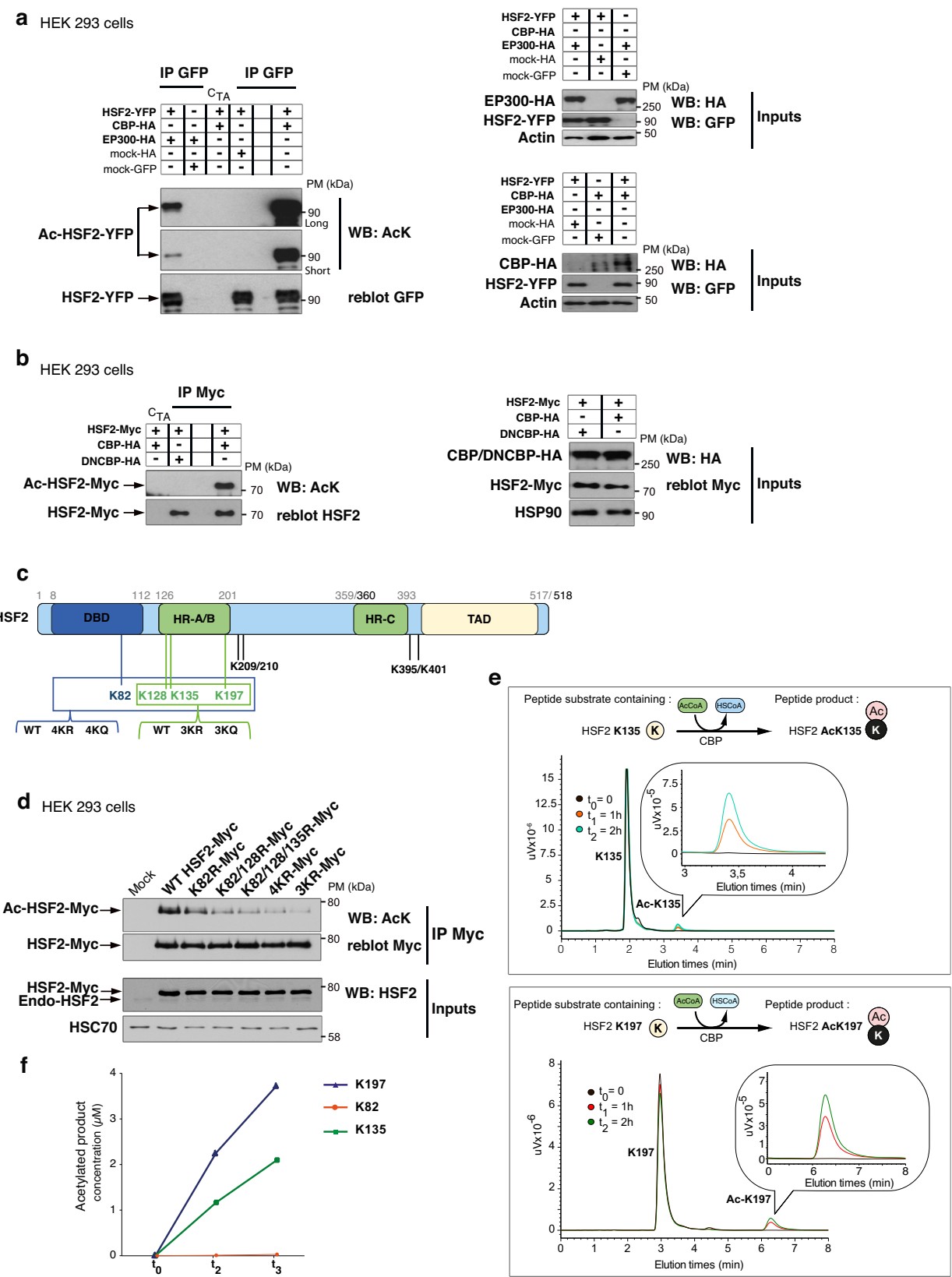

the core catalytic domain of CBP. The CBP-Full-HAT contains a Bromodomain (BD), a cysteine/histidine-rich region (CH2)—made of a zinc-finger containing RING domain and a plant homeodomain (PHD)[31]—and a HAT domain (Fig. 3a), and allows the coupling of substrate recognition and histone/lysine acetyltransferase activity (as in EP300)[32,33]. With biolayer interferometry, we observed that the

recombinant Full-HAT domain directly interacted with immobilized biotinylated recombinant full-length HSF2 (recHSF2) (Fig. 3b). Within this region, the recombinant PHD domain, but not the HAT, RING or BD domain, was able to interact with HSF2, in a similar manner as the Full-HAT domain (Fig. 3b). It is likely that the interaction between HSF2 and the catalytic HAT domain was too transient to be captured in these

**Fig. 2 | HSF2 is acetylated by CBP and EP300 in normal conditions. a** Representative immunoblots of immunoprecipitated HSF2-YFP (IP GFP) from HEK 293 cells transfected with combinations of tagged constructs, YFP-HSF2, HA-CBP, HA-EP300, mock-HA or mock-GFP, showing that ectopically expressed YFP-HSF2 is acetylated by exogenous HA-CBP or HA-EP300 ($n = 5$). C$_{TA}$, Trap®-A beads used as a negative control. Inputs, total proteins in input samples. Short and long, different exposure times. **b** Representative immunoblots of immunoprecipitated HSF2-Myc (IP Myc) from HEK 293 cells transfected with combinations of tagged constructs, HSF2-Myc, HA-CBP, or DNCBP (dominant-negative form of CBP) showing HSF2-Myc protein is acetylated by CBP but not by a dominant-negative form of CBP ($n = 2$). C$_{TA}$, Trap®-A beads used as a negative control. **c** Schematic representation of the eight main acetylated lysine residues of the HSF2 protein. DBD DNA-binding domain, HR-A/B hydrophobic heptad repeat, HR-C leucine-zipper-containing domain controlling oligomerization (TAD, activation domain). The numbers of the amino acids located at domain boundaries are indicated in gray (mouse HSF2) or black (human HSF2, if different). The four (blue box, K82, K128, K135, K197) or three (green box, K128, K135, K197) lysine residues in DBD, and/or HR-A/B were mutated into glutamines (4KQ, 3KQ) or arginines (4KR, 3KR). **d** Representative immunoblots of immunoprecipitated HSF2-Myc (IP Myc) from HEK 293 cells, co-transfected with EP300-HA and HSF2-Myc wild-type (WT) or HSF2-Myc carrying mutations on the indicated lysine residues showing that concommitant mutations of three or four lysine to arginine (3KR or 4KR) or glutamine (3KQ or 4KQ) residues decrease global HSF2 acetylation levels ($n = 3$). **e** Time course elution of HSF2K197 and HSF2K135 peptides detected by reverse phase-ultra-fast liquid chromatography (RP-UFLC) after 0 (black), 1 (red), or 2 (green) hours of acetylation by CBP-Full-HAT, monitored by fluorescence emission at 530 nm. uV arbitrary unit of fluorescence. See Methods for HSF2K197 and HSF2K135 peptide sequences and Supplementary Fig. 2. **f** Quantification of the in vitro acetylated HSF2 peptides containing K82, K135, and K197 residues detected by RP-UFLC. Source data are provided as a Source Data file.

experiments, since the HAT domain needs other CBP domains to interact with its substrates, including the PHD[34]. We determined that the K$_D$ of HSF2 interaction with the CBP-Full-HAT domain was 1.003E$^{-09}$ M (±2.343E$^{-11}$ R2 = 0.988488, Supplementary Fig. 3a). Our data strongly suggest that HSF2 is a bona fide substrate of CBP and potentially also of EP300, since their HAT domains display 86% identity.

### HSF2 interacts with CBP/EP300 via its oligomerization HR-A/B domain

The CBP/EP300 enzymes interact with their protein substrates through different anchorage domains, which subsequently allows the CBP/EP300 HAT domain to catalyze acetylation of these anchored substrates. Mutations in such anchorage domains can lead to pathological consequences[9]. We therefore dissected the mode of anchorage between CBP/EP300 and HSF2 proteins. We first confirmed their interaction by co-immunoprecipitation of tagged, exogenously expressed HSF2 and CBP/EP300 proteins, using GFP-trap assay (Supplementary Fig. 3b, c). We then validated in cellulo these interactions by the fluorescent three-hybrid assay, which allows to vizualise the interaction between these two proteins through their co-recruitment on a *LacOp* array locus (Fig. 3c)[35]. In this assay, HSF2-YFP was recruited by a GFP binder to the *LacOp* array locus (green spot) to which we observed that CBP-HA or EP300-HA was co-recruited (red spot) (Fig. 3d, e, Supplementary Fig. 3d, f, g). Furthermore, the abundance of CBP in BHK cells allowed us to detect the co-recruitment of HSF2-YFP with endogenous CBP (Fig. 3d, Supplementary Fig. 3e, g).

To determine which HSF2 domains were important for its interaction with CBP, we expressed Flag-HSF2 deletion mutants. We showed that the deletion of HR-A/B domain (but not DBD) led to a marked decrease in HSF2 acetylation (Supplementary Fig. 4a–c). These results were in line with our findings that the major acetylated lysine residues reside within the HR-A/B domain. The deletion of the transcription activation domain (TAD)[36] also resulted in decreased acetylation of Flag-HSF2 and was associated with decreased interaction with CBP (Supplementary Fig. 4a–d). Our data suggest that the HR-A/B oligomerization domain of HSF2, containing the key acetylated lysine residues, is involved, in a specific manner, in the interaction with CBP/EP300.

### CBP KIX domain binds to KIX motifs in the HSF2 HR-A/B domain

The anchorage of CBP/EP300 to many transcription factors occurs via different binding sites, including the kinase-inducible domain (KID) interacting domain (KIX domain) (Fig. 3a). This KIX domain contains two distinct binding sites that are able to recognize the "ΦXXΦΦ" KIX motif, where "Φ" is a hydrophobic residue, and "X" any amino acid[37–39]. We identified several conserved, overlapping, and juxtaposed KIX motifs in the HR-A/B domain of HSF2 (Fig. 4a). We modeled the interaction between the HSF2 HR-A/B KIX motifs and the CBP KIX

domain. Based on sequence similarities between the HR-A/B domain, lipoprotein Lpp56, and the transcription factors GCN4, ATF2, and PTRF (Supplementary Fig. 4e, Methods), we first developed a structural model of the HSF2 trimeric, triple-coiled coil, HR-A/B domain (Supplementary Fig. 4f). Second, we investigated the possibility of interactions of the KIX recognition motifs in the HR-A/B region of HSF2 with the CBP KIX domain. The best poses suggested that the HR-A/B KIX motif region contacted the so-called "c-Myb surface" within the CBP KIX domain[7], thereby proposing a close interaction of the HSF2 KIX motifs with the tyrosine residue Y650 of CBP (Fig. 4b, c and Supplementary Fig. 4g). We next examined the impact of in silico mutations of the K177, Q180, F181, V183 residues, which are present within the KIX motifs of HSF2 and involved in the contact with CBP (Fig. 4d). Either K177A or Q180A mutation within the HSF2 KIX motifs disrupted the HSF2-KIX domain interaction, in contrast to either F181A or V183A mutation (Supplementary Fig. 4g, Supplementary Table 2). Finally, we assessed the impact of in silico mutation of the Y650 amino acid of the CBP KIX domain, a residue mutated in RSTS patients[9]. The in silico mutation Y650A in CBP profoundly decreased the probability of interaction of the CBP KIX domain with the HSF2 KIX motifs (Fig. 4e, Supplementary Fig. 4g, Supplementary Table 2). Using recombinant proteins, we verified that HSF2 directly interacted with the CBP KIX domain in in vitro co-immunoprecipitation experiments and confirmed that the Y650A mutation disrupted HSF2 and KIX interaction (Fig. 4f). Thereby, we identify Y650 as a residue critical for interaction between the KIX domain of CBP and the KIX motifs located within the HSF2 oligomerization domain.

### The acetylation of HSF2 governs its stability

We explored the functional impact of the CBP/EP300-mediated acetylation of HSF2, in the neural cell line N2A (into which HSF2 and CBP/EP300 interact; Supplementary Fig. 5a). For this, we inhibited CBP/EP300 activity using the specific inhibitor C646[33,40]. The pharmacological inhibition of CBP/EP300 decreased the endogenous HSF2 protein levels, which was abolished by treatment with the proteasome inhibitor MG132 (Fig. 5a, Supplementary Fig. 5b), thereby providing evidence for acetylation affecting proteasomal-dependent HSF2 stability. To further investigate the role of acetylation in the regulation of HSF2 protein levels we transfected *HSF2*KO U2OS cells[23] (Supplementary Fig. 5c, d) with wild-type HSF2 or HSF2 acetylation mutants. These mutants mimic either constitutively acetylated (3KQ) or non-acetylated (3KR) HSF2 and are functional (Supplementary Fig. 5d–f). To monitor the decay of a pre-existing pool of HSF2 molecules, we performed pulse-chase experiments using the SNAP-TAG technology[41]. A pool of SNAP-HSF2 molecules was covalently labeled by adding a fluorescent substrate to the cells. At t$_0$, a blocking non-fluorescent substrate was added, quenching the incorporation of the fluorescent substrate to newly synthesized HSF2 molecules (Fig. 5b), allowing us to measure the decay in the fluorescence intensity of the corresponding

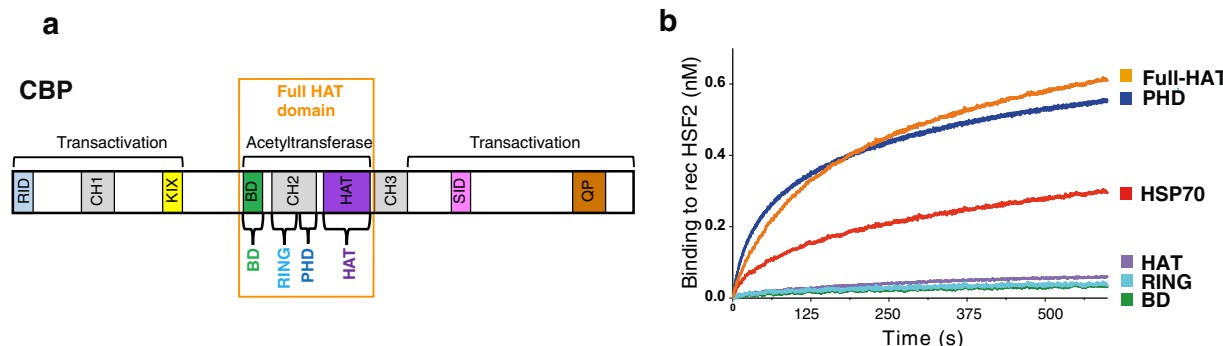

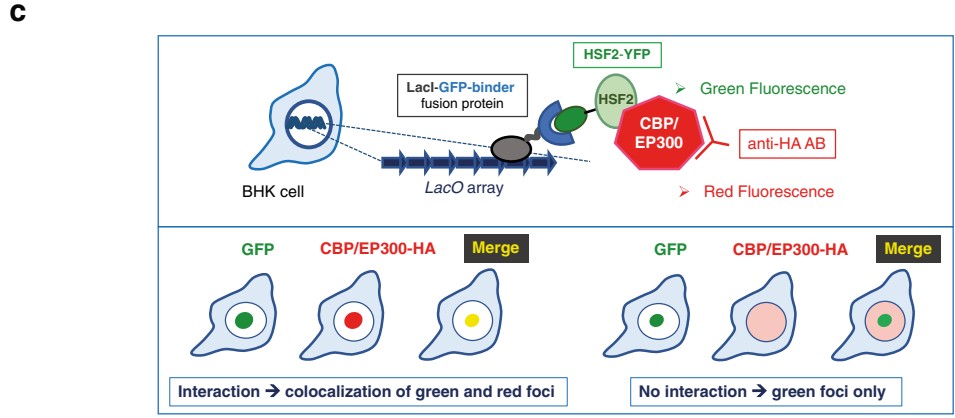

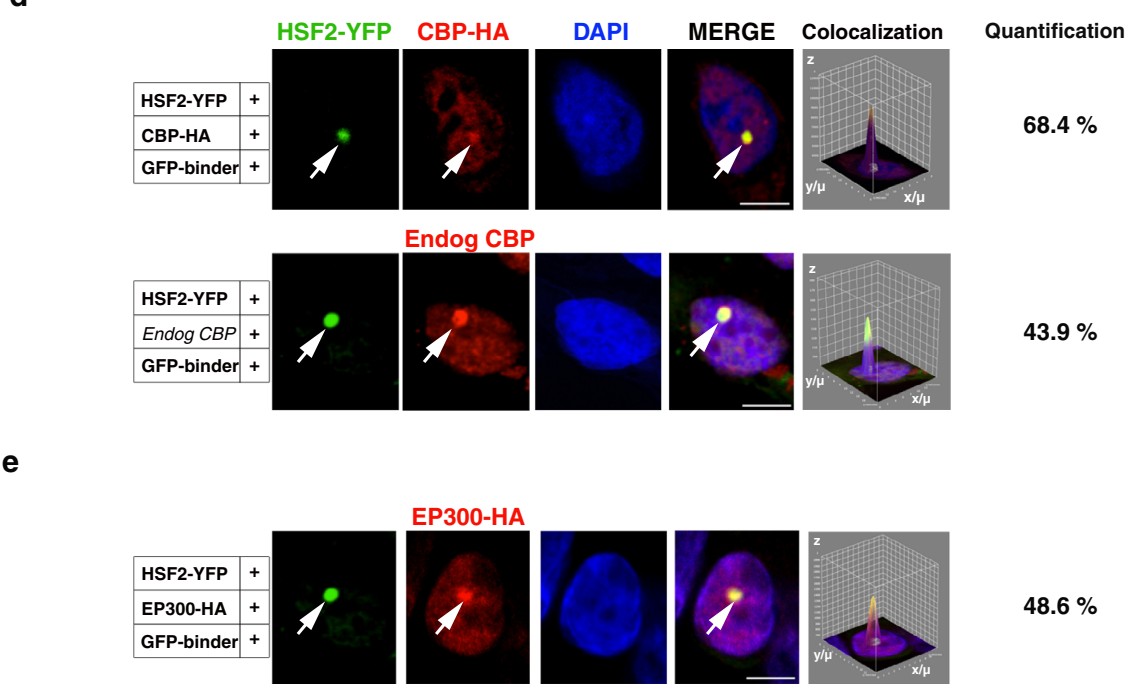

labeled HSF2 bands. When *HSF2*KO cells were transfected with wild-type SNAP-HSF2 (SNAP-HSF2 WT), a ~50% decay in fluorescence intensity was observed within 5 h (Fig. 5c, d). Preventing HSF2 acetylation (SNAP-HSF2 3KR) resulted in a similar decay (Fig. 5c, d). In contrast, mimicking acetylation with SNAP-HSF2 3KQ protected HSF2 from decay (Fig. 5c, d). Proteasome inhibition by MG132 prevented the

decrease in SNAP-HSF2 WT and 3KR fluorescent intensity, showing that SNAP-HSF2 decay is due to proteasomal degradation of the protein (Fig. 5e). Moreover, we observed that mimicking the acetylation of HSF2 by expressing Myc-HSF2 3KQ limited the poly-ubiquitination of HSF2, when compared to HEK 293 cells expressing either WT or 3KR HSF2 (Fig. 5f). Because heat shock provokes the degradation of the

**Fig. 3 | HSF2 interacts with CBP and EP300 in normal conditions. a** Schematic representation of CBP protein domains. The ability of CBP to bind a very large number of proteins is mediated by several conserved protein binding domains, including the nuclear receptor interaction domain (RID), the cysteine/histidine-rich region 1 (CH1), the KIX domain, the bromodomain (BD), the CH2 containing a PHD and a RING domain, the HAT, the CH3, the steroid receptor co-activator-1 interaction domain (SID) and the glutamine- and proline-rich domain (QP)[33].
**b** Representative kinetics of recombinant HSF2 binding to His-tagged CBP domains or HSP70 (positive control) by biolayer interferometry (*n* = 3). **c** Schematic representation of the principle of the fluorescent-3-hybrid (F3H) assay. Genomic integration of a *LacO* array allows the focal recruitment in the nucleus of a LacI fused to the GFP binder, which in turn recruits the GFP-tagged probe (HSF2-YFP) and its

potential interactants (CBP/EP300), being either endogenous or brought by overexpression. **d, e** Representative confocal sections of BHK cells carrying a stably integrated Lac-operator array, transfected with LacI-GFP binder, HSF2-YFP, and CBP-HA (**d**) or EP300-HA (**e**) showing the interaction between HSF2-YFP (green) and exogenous CBP-HA, endogenous CBP (**d**) or with exogenous EP300-HA (**e**) (red). White arrows, co-localization of HSF2 and CBP or EP300 at the *LacO* array. Negative controls are shown in Supplementary Fig. 3d–g. Scale bar: 10 μm. Graphs represent the combined signal intensity of the two fluorescence signals at the *LacO* array. Quantification: percentage of cells showing co-recruitment of YFP-HSF2 and EP300-HA, CBP-HA or endogenous CBP at the *LacO* array (*n* = 3 or 4, average of 100 counted cells per experiment). Source data are provided as a Source Data file.

HSF2[21], we next analyzed the impact of acetylation on the heat-shock-induced decay of HSF2. We showed that mimicking HSF2 acetylation, using SNAP-HSF2 3KQ, mitigated the decay of fluorescence intensity induced by heat shock, when compared to SNAP-HSF2 WT or 3KR (Fig. 5g). Altogether these experiments demonstrate that HSF2 acetylation prevents the proteasomal degradation of HSF2 both under non-stress and stress conditions.

### Reduced HSF2 protein levels in Rubinstein-Taybi syndrome

Because the HSF pathway is involved both in stress reponses and neurodevelopment, and destabilized in the presence of mutated CBP or EP300, it could directly participate in the pathology of RSTS. To determine the functional impact of impaired CBP and EP300 activities on the levels of the HSF2 protein in this pathological context, we first compared the amounts of HSF2 protein in cells derived from either healthy donors (HD) or RSTS patients carrying mutations or deletions either in the *CBP* or *EP300* genes (Fig. 6a; Methods). We used RSTS patients-derived human primary skin fibroblasts (hPSFs), in which the effect of CBP/EP300 mutations were validated by the observation of a reduced amount of acetylated lysine residue K27 in histone H3 (AcH3K27), when compared to HD (Supplementary Fig. 6a). This included an RSTS$_{CBP}$ patient P1 (RSTS$_{CBP}$ P1) carrying a single-point mutation in the catalytic HAT domain of CBP and an RSTS$_{EP300}$ patient P2 (RSTS P2$_{EP300}$) carrying a deletion in the KIX domain of EP300, two domains important for interaction with and acetylation of HSF2. We observed that HSF2 protein levels were markedly decreased in hPSFs from both RSTS patients (Fig. 6a–c, Supplementary Fig. 6b). HSF2 levels were restored to comparable levels of HD when these hPSFs were treated with the proteasome inhibitor MG132 (Fig. 6b, c); this was not the case upon inhibition of Class I HDACs (Fig. 6b, c, Supplementary Fig. 6c, d). This finding suggests that both the catalytic HAT domain (mutated in the RSTS P1$_{CBP}$) and the KIX domain (largely deleted in the RSTS P2$_{EP300}$) are required for the regulation of HSF2 stability, in line with our in silico results. Overall, these data show that the proteasomal turnover of HSF2 is increased in RSTS hPSFs and that CBP and EP300 are key regulators of HSF2 protein stability in this pathological context.

### Impaired heat-shock response in RSTS primary cells

Although HSF1 is the essential driver of the acute heat-shock response in mammals[18], HSF2 acts as a fine tuner of the heat-shock response and determines the magnitude to which the applied heat stress induces *HSP* gene expression[42]. Therefore, we examined the ability of RSTS cells to mount the heat-shock response. In the absence of heat stress, we observed that RSTS hPSFs displayed lower amounts of HSP70 and HSP90 than their HD counterparts (Fig. 6d). Furthermore, RSTS hPSFs exhibited limited capacity to accumulate HSP70 upon acute heat shock and during the recovery phase from heat stress (Fig. 6d). Importantly, this limited induction did not result from impairment of HSF1 activation, since HSF1 was activated by heat shock in RSTS hPSFs, as assessed by its slowed mobility shift in SDS-PAGE (Fig. 6d). This shift

is a hallmark of HSF1 hyperphosphorylation, which, although not required for HSF1 activation, accompanies the induction of HSF1 transactivation potential[43,44]. As mentioned above, HSF1 and HSF2 do not only control the transcription of the *HSP* genes in response to acute heat stress, but they also upregulate the transcription of *Sat III* 9q12 heterochromatin regions, in which nuclear stress bodies (nSBs) are formed[45]. We therefore used nSBs as a read-out for assessing the heat-shock response in RSTS cells and observed that the stress-inducible formation of nSBs was reduced by more than 50% in RSTS$_{EP300}$ hPSFs when compared to their HD counterparts at 42 °C or 43 °C (Fig. 6e, Supplementary Fig. 6f). A similar reduction in the formation of nSBs was observed in RSTS lymphoblastoid cells derived from three other patients (Supplementary Fig. 6g).

To decipher the role of the limited amounts of HSF2 protein in the impairment of RSTS hPSFs to form nSBs in response to heat shock, we rescued the levels of HSF2 by transfecting RSTS hPSFs with an HSF2 3KQ construct. The ability of RSTS hPSFs to form nSBs in response to heat shock was restored to comparable levels as those observed in HD cells (Fig. 7a, Supplementary Fig. 7, Supplementary Fig. 8a). In summary, these data suggest that the lack of HSF2 in RSTS cells results in their altered capacities to activate a major stress-responsive pathway.

### Impaired expression of HSF2 target genes in RSTS cells

Because HSF2 is involved in the tight control of neurodevelopmental gene expression[14,15,23], we investigated whether the lack of HSF2 prevented the proper expression of neurodevelopmental genes in RSTS cells, which could contribute to the neurodevelopmental features of this pathology. We examined two HSF2 target genes that are expressed in hPSFs and whose expression is dependent on HSF2, in mouse fetal cortices: *HSPH1/HSP110* and *N-cadherin* (Supplementary Fig. 8b). The promoter region of the *HSPH1/HSP110* gene is bound by HSF2[46], and the HSP110 protein is involved in brain integrity in models of brain trauma, neuropsychiatric, and neurodegenerative disorders[47–49], as well as in important neurodevelopmental processes[50,51]. N-cadherin plays important roles in brain formation and integrity[25,27,52], and its expression was recently shown to be tightly controlled by HSF2[23]. We found that both HSP110 and N-cadherin protein levels were decreased in RSTS patient hPSFs, in comparison to HDs (Fig. 7b). Moreover, we observed a clearly diminished N-cadherin labeling at cell–cell junctions in RSTS hPSFs, when compared to HDs (Fig. 7c, Supplementary Fig. 8c). These results indicate that the function of N-cadherin might be compromised in RSTS cells, due to the reduction in HSF2 levels. We therefore tested whether treatment with bortezomib (BTZ), a proteasome inhibitor used as a clinical cancer drug, could, at subthreshold doses (5–10 nM), restore HSP110 and N-cadherin levels in RSTS cells. Indeed, we and others have demonstrated that the amount of HSF2 protein is increased by low doses of BTZ (≤10 nM in less than 20 h)[23,53,54]. We found that the upregulation of HSF2 levels by subthreshold doses of BTZ (5–10 nM) was accompanied by restoration of N-cadherin levels and increase in HSP110 expression (Fig. 7d).

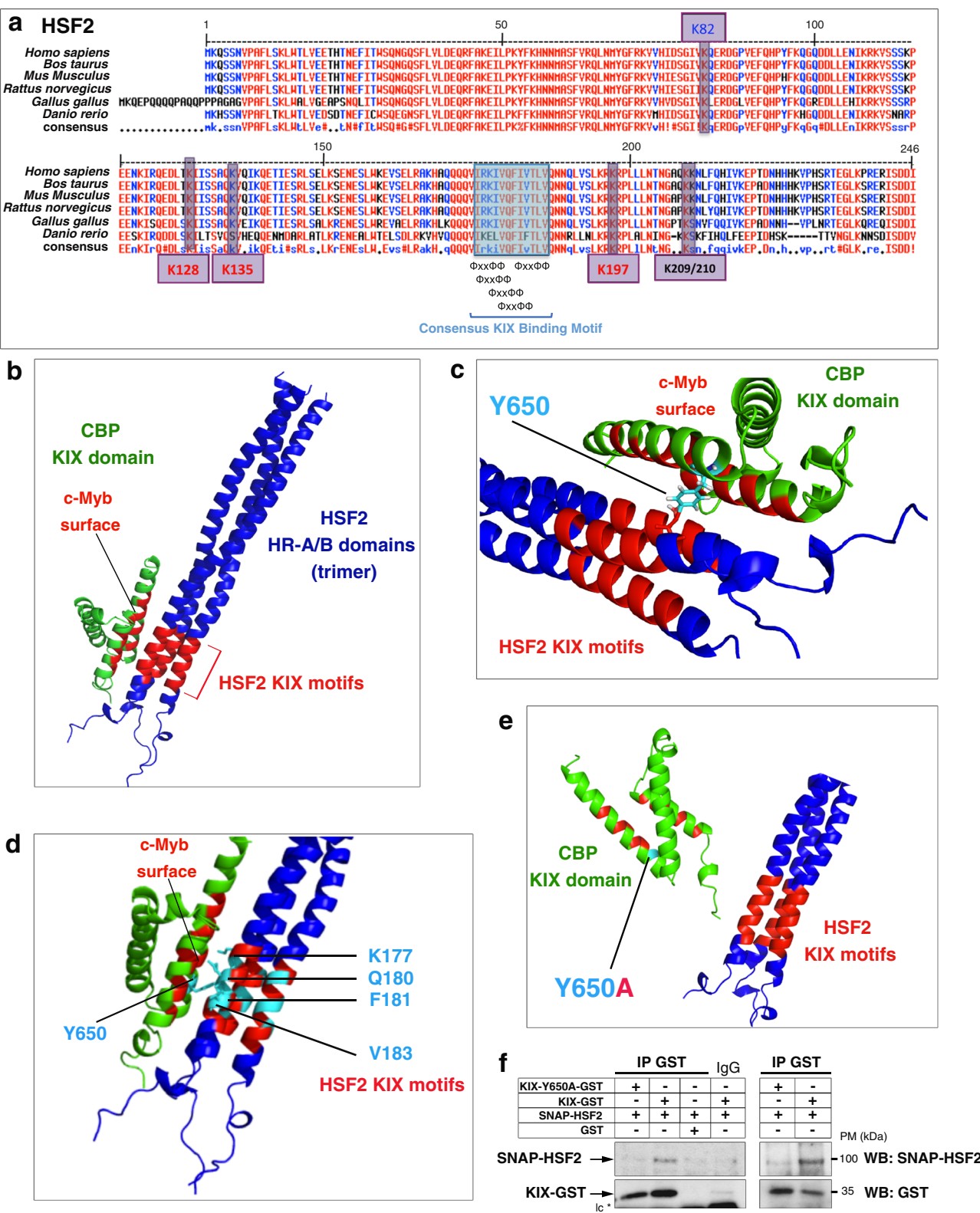

In conclusion, these results highlight the functional impact of the dysregulation of HSF2 stability, as induced by CBP and EP300 mutations in the RSTS context, leading to the downregulation of genes that govern a major stress-responsive pathway and/or control neurodevelopment under non-stress conditions. These molecular consequences of CBP/EP300-dependent HSF2 dysregulation have a potential to contribute to the RSTS pathology, as discussed below.

### Dysregulated CBP/EP300-HSF2-N-cadherin cascade in RSTS iNPCs/hCOs

To investigate whether the regulation of HSF2 stability by CBP/EP300 was active in neurodevelopmental contexts, we derived iPSC clones from RSTS P1$_{CBP}$ and RSTS P2$_{EP300}$ hPSFs (Supplementary Fig. 9) and differentiated them into iNPCs in 2D-cultures (Supplementary Fig. 10a, b) or in hCOs 3D-cultures (Supplementary Fig. 10c). As a control, we generated iNPCs and hCOs from the IMR90-4 iPSC line, which was

**Fig. 4 | Modeling of CBP and HSF2 interaction. a** Amino acid sequence of the KIX-binding motifs located in the HSF2 HR-A/B region. Blue rectangle, conserved KIX-binding motif sequences ("ΦΧΧΦΦ"). Purple boxes, positions of the very conserved and major acetylated lysine residues; K82 (blue) is located in the DBD, K128, K135, and K197 (red) are located in the HR-A/B and K209/K210 (black) is located downstream the HR-A/B. **b** In silico model structure of the CBP KIX domain and the HSF2 HR-A/B domain interaction. Representation of the HSF2 HR-A/B domains In the HSF2 trimer, as a triple-coiled coil (in blue). The KIX recognition motifs of HSF2 are indicated in red. Representation of the KIX domain of CBP, a triple helical globular domain (in green). The c-Myb surface of the KIX domain is indicated in red. **c** In silico model. Magnification of the HSF2 and CBP interaction domains shown in **b** showing the tyrosine residue Y650 (pale blue) within the c-Myb surface of the CBP KIX domain in contact with the KIX recognition motifs of the HSF2 HR-A/B domain. **d** In silico model representation of the position of the four residues of HSF2 KIX recognition motifs and of Y650 of the CBP KIX domain that have been analyzed by in silico mutation. **e** In silico Y650A mutation disrupts interaction between the HSF2 KIX motifs and the CBP KIX domain (Firedock analysis). **f** Representative immunoblots of immunoprecipitated CBP KIX domain (IP GST) after in vitro interaction experiments between wild-type or mutated CBP KIX-GST and SNAP-HSF2 recombinant proteins produced in bacteria and reticulocyte lysates showing Y650A mutation disrupts interaction between the HSF2 KIX motifs and the CBP KIX domain (*n* = 3). lc IgG light chain. The left and right immunoblots correspond to two independent experiments. Source data are provided as a Source Data file.

derived from a heathly donor and has been used to produce bona fide hCOs (HD iNPCs and HD hCOs)[55]. Because the *Hsf2*KO mice show early signs of neurodevelopmental defects, especially in the number and organization of neuroprogenitor cells (e.g., radial glia cells)[14], we focused on iNPCs, as well as on early-stage hCOs (D5 to D25).

First, we showed that both RSTS$_{EP300}$iNPCs (Fig. 7e, f, Supplementary Fig. 10d) and RSTS$_{EP300}$ and RSTS$_{CBP}$ hCOs, (Fig. 7g, Supplementary Fig. 10e) reproducibly exhibited lower HSF2, N-cadherin, and HSP110 levels than HD counterparts, suggesting that CBP/EP300 mutations compromised HSF2 levels and the expression of its targets, not only in RSTS hPSFs, but also in these models of neurodevelopment. Next, we addressed whether the stabilization of the HSF2 protein in RSTS iNPCs and hCOs could lead to increase in N-cadherin and HSP110 levels. As for RSTS hPSFs (Figs. 6b, c and 7d), we treated RSTS iNPCs and hCOs with BTZ or MG132 to pharmacologically modulate the activity of the HSF2 pathway. Surprisingly, BTZ failed to increase HSF2 protein amounts in these neural systems, although it increases HSF2 levels at higher levels than MG132 in other cell systems[23,54], including hPSFs (Figs. 6b, c and 7d, Supplementary Discussion). In contrast, MG132 led to increased HSF2 levels, as expected. This increase was also accompanied by elevation of N-cadherin and HSP110 levels, in both RSTS iNPCs and hCOs (Fig. 8a, b, Supplementary Fig. 11a, b). We verified that the doses of MG132 that were used to stabilize HSF2 neither interferred with iNPC neural differentiation state, nor hCO differentiation (Supplementary Fig. 11a, d), nor globally disturbed the proteome, as suggested by unaffected levels of PCNA, a well-known substrate of the proteasome activity (Supplementary Fig. 11c, e). We performed two control experiments to confirm that the effect of MG132 on the elevation of N-cadherin and HSP110 levels was not due to direct stabilization of these proteins, but occurred via increase in HSF2 levels. First, we showed that no increase in N-cadherin and HSP110 levels was observed in HD hCOs upon MG132 treatments, while global levels of protein ubiquitination were equivalent in HD compared to RSTS hCOs (Supplementary Fig. 11f, g). Second, we examined NDE1, an important player in cortical progenitor division[56], whose gene is bound by HSF2[15], but whose expression is not significantly regulated by HSF2 in the mouse developing cortex, in contrast to N-cadherin and HSPs. Importantly, NDE1 protein levels did not exhibit major changes in response to MG132 exposure in the RSTS neural context (Fig. 8b, Supplementary Fig. 11b). This suggests that, at these doses, MG132 does not globally stabilize HSF2 targets, neither in HD, nor in RSTS neural models. Consequently, the elevation in HSP110 and N-cadherin levels upon MG132 treatment in RSTS models likely occurs through increased levels of their strong positive regulator HSF2. Together with our earlier study uncovering HSF2 as a major regulator of N-cadherin[23], our data suggest that the CBP/EP300-HSF2-N-cadherin cascade, is not only active in RSTS primary cells (Fig. 7b–d), but also in neurodevelopmental contexts.

In line with the finding that diminished N-cadherin levels were likely due to the dysregulation of HSF2 in RSTS iNPCs and hCOs, we identified altered neurodevelopmental characteristics in RSTS-derived iNPCs and hCOs, in comparison to HD counterparts. Cell–cell adhesion, partly through the formation of apical adherens junctions involving N-cadherin and its partners, is a crucial process during neurodevelopment, which governs, in particular, the radial orientation of neuroprogenitor cells and their division process, including the specific positioning of mitoses at the apical side of the germinal zone[57,58]. Accordingly, although we were able to produce iNPCs and hCOs from RSTS iPSCs (Supplementary Fig. 10), deeper analyses revealed that these models display defects coherent with both mouse cortical *Hsf2*$^{-/-}$ phenotypes[14] and cell–cell adhesion deficits. First, we observed perturbations in the radial organization of RSTS$_{EP300}$ iNPCs. Indeed, HD iNPCs formed rosette-like structures in 2D-cultures, whose radial organization, centered around groups of PAX6 + cells, was visualized by staining radial glia-like cells for FABP7 (Fig. 8c). In contrast, such radial organization of FABP7 + radial glia-like cells could not be observed in RSTS iNPC cultures. Rather, PAX6 + cells appeared spread out, i.e., not organized in clusters surrounded by FABP7 + cells (Fig. 8c). This echoes the phenotype of mouse *Hsf2*$^{-/-}$ cortices, which display perturbation of radial glia fiber organization (*Hsf2*$^{tm1Mmr}$ mouse strain)[14]. Second, by staining apical mitoses, using a phospho-histone H3 marker (H3S10Ph), we demonstrated that RSTS$_{EP300}$ hCOs displayed a higher rate of ectopic mitoses, located at distance from the apical zone of the loops (stained by ZO-1), in a statistically significant manner, compared to HD hCOs (Fig. 8d, Supplementary Fig. 11h).

In conclusion, we show that the pathway that we have unraveled, which links the regulation of HSF2 stability to CBP/EP300, is recapitulated in 2D and 3D models of human neurodevelopment, and that its functional relevance relies on the control of N-cadherin expression, keeping in with phenotypical traits linked to cell–cell adhesion defects observed in the pathological CBP/EP300-deficient RSTS1 and RSTS2 contexts.

## Discussion (see also Supplementary Discussion)

We identify the HSF2 pathway, a central stress-responsive pathway that also controls brain development in physiological conditions, as being disrupted in RSTS patients. The KATs CBP and EP300 catalyze the acetylation of HSF2, thereby contributing to its stability in hPSFs and in neural models derived from patient cells. Restoring HSF2 levels, either pharmacologically or genetically, restores the impaired stress response and expression of neurodevelopmental genes, in particular N-cadherin. Finally, we show that the dysregulation of the CBP/EP300/HSF2-N-cadherin cascade is recapitulated in RSTS1 and RSTS2 patient-derived iNPCs and hCOs, and rescued by stabilization of HSF2. The occurrence of phenotypic traits reminiscent of the HSF2-deficient mouse cortex and characteristic alteration of cell–cell adhesion in the neurogenic niche in RSTS hCOs are suggestive of pivotal role for HSF2 in RSTS neural pathology.

We show that the full-length HSF2 protein interacts with the CBP core catalytic domain in vitro, confirming that HSF2 is a bona fide substrate of CBP. This interaction might have a profound impact on HSF2 stability, contributing to the pathology in a number of RSTS patients. Indeed, key residues or domains (KIX, PHD, HAT), when mutated in CBP/EP300 RSTS patients, compromise their interaction

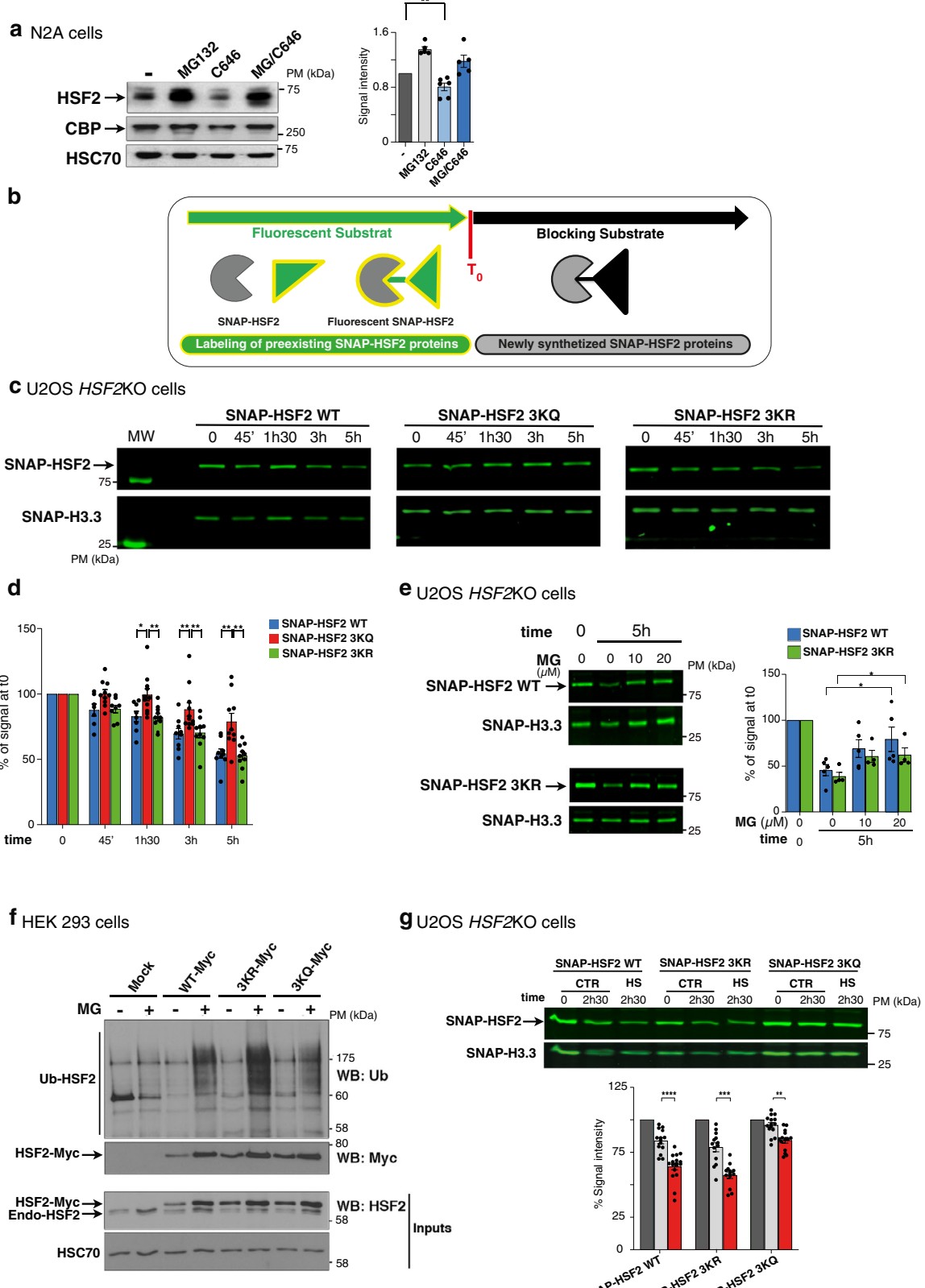

with HSF2 and/or its acetylation. HSF2 interacts with the PHD domain, which is part of the catalytic core of the CBP/EP300 proteins[32,34] and is mutated in some RSTS patients[59]. We find that the HSF2 HR-A/B oligomerization domain (containing KIX-binding motifs), but not the DNA-binding domain, is necessary for the anchoring interaction of HSF2 with CBP KIX domain, as well as for HSF2 acetylation.

Furthermore, the tyrosine residue Y650 in the CBP KIX domain is critical for this interaction and its mutation is associated with a severe RSTS neurodevelopmental phenotype[9]. The KIX domain contains two binding sites, the Mixed Lineage Leukemia protein (MLL) site and the c-Myb site[7,60], the latter being the one bound by the HR-A/B KIX motifs of HSF2. The cooperative binding of two transcription factors on these

**Fig. 5 | Impact of preventing or mimicking acetylation of lysine residues K128, K135, and K197 on HSF2 protein stability. a** Representative immunoblots. The inhibition of CBP/EP300 decreases HSF2 and is counteracted by proteasome inhibition in N2A cells treated with the CBP/EP300 inhibitor C646 (40 µM, 4 h) and/or with MG132 (20 µM, 6 h) (*n* = 4). Quantification of HSF2 signal intensity, normalized by HSC70 and relative to vehicle-treated samples (−). Error bars, mean ± standard error of the mean (SEM), *$p$ = 0.0022. **b** Scheme of the principle of SNAP-TAG pulse-chase experiments. **c** Representative electrophoresis images of protein extracts from *HSF2*KO or WT U2OS cells expressing SNAP-tagged WT, 3KQ or 3KR HSF2, SNAP-labeled, and showing the decay of 3KQ HSF2 mutant protein levels (0–5 h). SNAP-H3.3, loading control. **d** Quantification of the fluorescent signal normalized to H3.3 and relative to the signal at $t_0$ (*n* = 7 with replicates). Error bars, mean ± SEM, *p* = 0.0112 (1h30,KQ vs.WT), *p* = 0.0029 (1h30,KR vs.KQ), *p* = 0.0045 (3 h,KQ vs.WT), *p* = 0.0039 (3 h, KQ vs. KR), *p* = 0.0076 (5 h, KQ vs. WT); *p* = 0.0015 (3 h, KQ vs.KR) *$p$ < 0.05; **$p$ < 0.01. **e** Representative electrophoresis images of protein extracts from *Hsf2*KO U2OS cells expressing SNAP-tagged HSF2 WT or 3KR, pretreated by

MG132 (vehicle (0), 10, 20 µM, 5 h), SNAP-labeled, and analyzed after 5 h, showing the decrease in SNAP-HSF2 WT and 3KR protein levels depending on proteasome activity (*n* = 3). Error bars, mean ± SEM. *p* = 0.0286 (KR vs. KR_MG), *p* = 0.0159 (WT vs. WT_MG) *$p$ < 0.05, quantification as in **d**. **f** Representative immunoblots of immunoprecipitated HSF2-Myc (IP Myc) from HEK 293 cells transfected with HSF2-Myc WT, 3KR, or 3KQ, and treated (+) or not (−) with MG132 (20 µM, 6 h), showing preferential poly-ubiquitination of the HSF2 3KR mutant protein, compared to HSF2 WT or 3KQ (*n* = 3). **g** Representative electrophoresis image as in **c**, but *Hsf2*KO U2OS cells were treated with HS (42 °C) or not, (CTR), and analyzed prior (0, grey) or after 2.5 h (light grey) of HS (red), showing increased HSF2 3KQ stability, upon HS, compared to WT or 3KR (*n* = 7). Error bars, mean ± SEM *** *p* = 0.0011 (3KQ, HS 2h30 vs. CTR 2h30)), ****$p$ < 0.0001 (WT, HS 2h30 vs. CTR 2h30); *p* = 0.0001 (3KR, HS2h30 vs. CTR 2h30). Quantification as in **d**. Significance was calculated by two-sided Mann−Whitney test in panels **a**, **d**, **e**, **g**. Source data are provided as a Source Data file.

two surfaces can potentially and mutually modulate each other binding. Therefore, the impairment of CBP/EP300 binding to HSF2 might also have functional and detrimental effects on other transcription factor pathways, broadening the manner by which HSF2 could contribute to the pathology.

Based on our data, the acetylation of HSF2 by CBP/EP300 limits its proteasomal degradation, a process which has been observed for other transcription factors, such as p53[61]. Moreover, the acetylation of the lysine residues K128, K135, and K197 of HSF2 limits its degradation both in unstressed and stressed conditions. This does not seem to occur by directly preventing their poly-ubiquitination. Indeed, only combined mutations of these lysines to glutamines (3KQ, mimicking HSF2 acetylation), but not to arginines (3KR), prevent HSF2 proteasomal degradation. In addition, 3KQ mutation decreases HSF2 poly-ubiquitination, whereas 3KR does not. Previous proteome-wide quantitative analyses of the ubiquitin-modified protein have revealed that the ubiquitination of HSF2 occurs at multiple residues spanning over the HSF2 protein, in addition to K128, K135, and K197. Most of these sites reside in the HR-A/B domain or its vicinity, suggesting a crosstalk between acetylation and ubiquitination (www.phosphosite.org)[18,62–64].

We find that the stress reponse is impaired in RSTS primary cells, compromising the normal induction of HSPs and the formation of nSBs in response to heat shock. The dysregulation of the HSF pathway is a shared feature among derived cells from five patients of diverse genetic origins. We can rescue this phenotype by introducing the stabilized form of HSF2, HSF2 3KQ, in RSTS cells, leading to the restoration of the ability of RSTS cells to mount a proper stress response. This result further supports the role of HSF2 in controlling the stress response in RSTS cells, in line with its role in the heat-shock response in other cell systems[42,45]. Importantly, we also find that the constitutive levels of HSPs in unstressed RSTS cells are diminished. We thus hypothetize that the alteration of the repertoire of chaperones might be causal to some RSTS pathological aspects and that manipulating HSF2 levels might open therapeutic perspectives.

We show that the endogenous HSF2 protein is acetylated in mouse neural embryonic tissues, in hCOs and in a neural cell line. The physiological HSF2 acetylation suggests that the acetylation might be a key regulatory event involved in the abundant expression and the role of HSF2 in cortical development. The functional importance of this regulation in neurodevelopmental pathology is underlined by the decreased expression of genes important for neurodevelopment, concommittant with HSF2 diminished levels in RSTS patient-derived primary and neural 2D and 3D models, and by its amelioration upon restoring HSF2 by proteasomal inhibition. In RSTS iNPCs, disturbances of radial, "rosette-like" iNPC organization are suggestive of defective cell–cell contacts, as is the presence of ectopic mitoses at early stages of hCO differentiation. Because N-cadherin expression is strongly regulated by HSF2[23] and is a major contributor in the formation of

apically localized adherens junctions in the neurogenic niche[57,58], it is likely that the alteration of the CBP/EP300-HSF2-N-cadherin cascade in RSTS contributes, at least partially, to these defects. These cell–cell adhesion defects are susceptible to lead to imbalance between proliferation and neuronal differentiation in RSTS models, as was shown in mouse and cortical organoid models into which the N-cadherin pathways is impaired[57,65–67]. In addition, the N-cadherin pathway has important roles in later stages of neurodevelopment, including synapse formation and plasticity[24,25,27] and, thus, has the potential of participating in many neurodevelopmental defects in RSTS patients. HSP110, also, is involved in synapse and adult brain integrity[48,49] and is critical for brain development, in particular via the control of the mitotic spindle and of the Wnt pathway[50,51]. N-cadherin also impacts the Wnt pathway[67], seemingly combining the action between these two HSF2 targets. Similarly, combined impacts of N-cadherin, N-cadherin partners[68], and other HSF2 target genes involved in cell adhesion, including many HSF2 targets of the cadherin superfamily[23], could participate in the RSTS pathology. In support of this idea, transcriptomic disturbances were identified in RSTS iPSC-derived neurons; affecting genes involved in cell polarity and adhesive functions, with impact in preterminal neuronal differentiation[69]. Finally, the various control experiments that we performed strongly suggest that the pharmacological rescue that we used is rather specific to HSF2 in terms of the restoration of N-cadherin and HSP110 expression. However, a clear proof of the causality of the defects in the CBP/EP300-HSF2-N-cadherin on these phenotypes would require a long-term rescue of HSF2, based, for example, on genome-editing strategies.

We have shown, for the first time to our knowledge, that the alteration of the stress-responsive HSF2 pathway, notably through the impairment of the CBP/EP300-HSF2-N-cadherin cascade, defines a critical aspect of this RSTS model. Our findings pave the way for future studies aiming at further deciphering the mechanisms regulated by this stress-responsive pathway for the understanding of neurodevelopmental deficiency observed in NDDs. In support of our findings, the recent reporting of a deleterious de novo mutation of the *HSF2* gene linked to Angelman Syndrome[70] is a further proof of the importance of the integrity of the HSF2 pathway in neurodevelopment contexts.

## Methods
### Ethical regulations
Our research complies with all relevant ethical regulations for the boards/committees and institutions that approved the study protocols.

For RSTS patients P1−P5, biopsies were performed in childhood, preadolescence or young adulthood, and the ratio of the number of males to the number of females was 3 to 2.

For the derivation of human iPSC lines from RSTS patients P1 (*CREBBP*) and RSTS patient P2 (*EP300*), skin biopsies were obtained at Hôpital Robert Debré (Paris, Assistance Publique-Hôpitaux de Paris

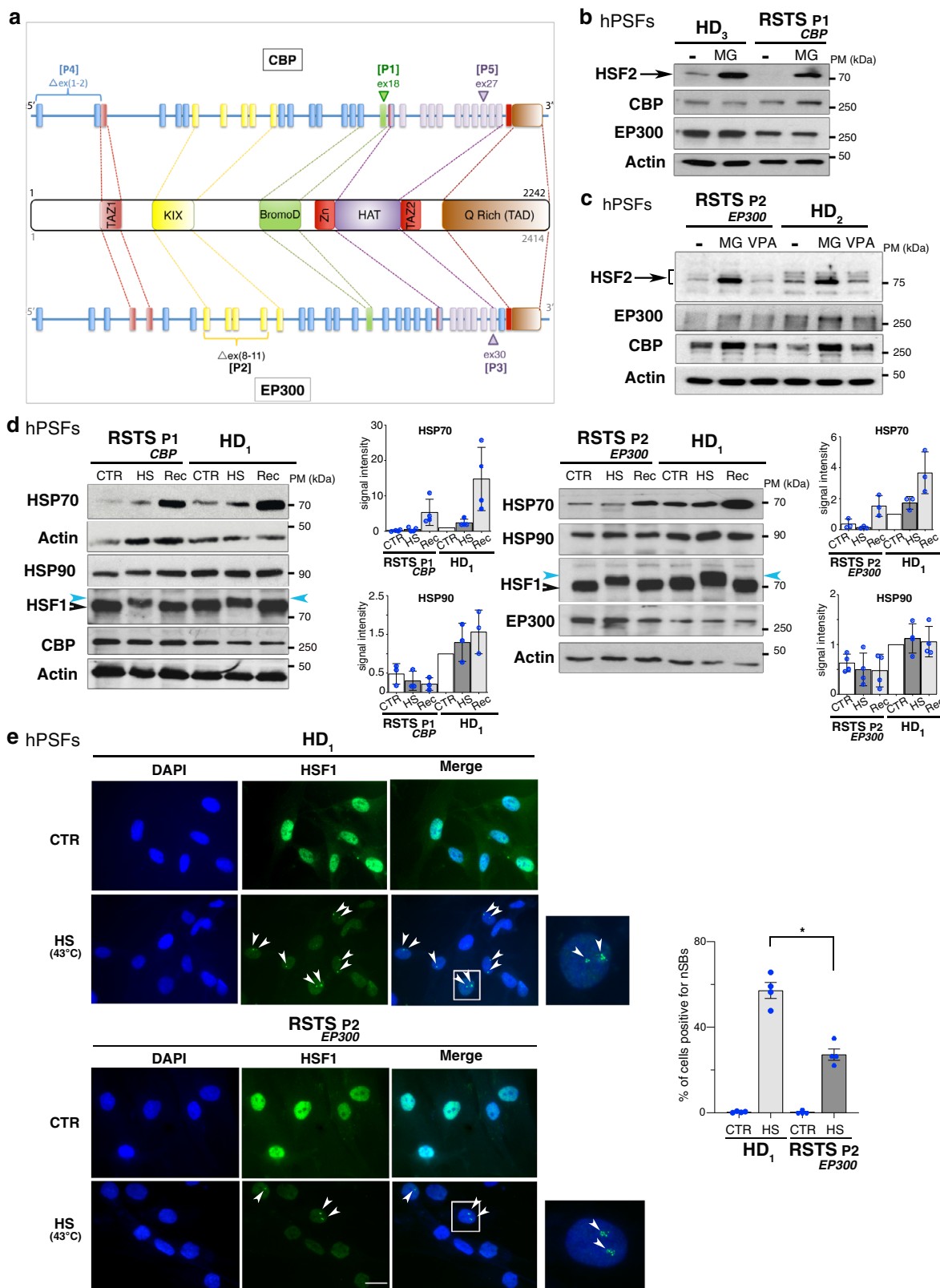

(AP-HP)). The corresponding research projects were approved by the National Ethics Committee (Comité de Protection des Personnes (CPP) Ile-de-France II, number 2010AO1481-38). Written informed consent for procurement of skin biopsies and use of these and cell lines derived from these within this study were obtained from the patients' legal guardians.

RSTS PSFs were derived from the patient RSTS P1 and P2 biopsies and belong to a collection of skin fibroblasts for in vitro culture, derived from patients with rare hereditary diseases (Biobank agreement n° P100128; « Fibroblastes en culture issus de peau de patients atteints de maladies héréditaires rares »), and located at the Centre Hospitalo-Universitaire Bicêtre (AP-HP). The derivation of primary skin

**Fig. 6 | Altered HSF2 protein levels and dysregulated stress response in cells from RSTS patients. a** Schematic representation of the mutations or deletions present in RSTS patients. The scheme of the genomic organization of the genes are taken from NCBI data base (NM_001429.3) (see Supplementary methods). **b** Representative immunoblots of protein extracts from HD and RSTS P1$_{CBP}$ hPSFs treated with 20 μM MG132 (6 h) showing reduced HSF2 levels in RSTS hPSFs, compared to HD, but restored levels in the presence of the proteasome inhibitor MG132 ($n = 3$). **c** Representative immunoblots of protein extracts from HD and RSTS P2$_{EP300}$ hPSFs treated with 20 μM MG132 (6 h) or 1 mM of the HDAC inhibitor VPA (3 h) showing that reduced HSF2 levels observed in RSTS hPSFs, compared to HD, are restored in the presence of the proteasome inhibitor MG132, while the HDAC inhibitor VPA does not restore HSF2 levels ($n = 3$). **d** Representative immunoblots of protein extracts from HD and RSTS P1$_{CBP}$ hPSFs in control (CTR), heat shock (HS, 1 h

at 42 °C), and recovery conditions (Rec, HS + 2 h at 37 °C) showing reduced HSP basal levels and induction by HS in RSTS, compared to HD hPSFs ($n = 3$). Blue arrowhead, hyperphosphorylated and thereby shifted HSF1 band. Quantification of HSP70 and HSP90 signal intensity in immunoblots, normalized to actin. Error bars, mean ± s.d. **e** Representative immunofluorescence of protein extracts from HD and RSTS P2$_{EP300}$ hPSFs in control (CTR) or heat-shock conditions (HS, 1 h at 43 °C), showing altered formation of nSBs (HSF1 nuclear speckles, green) upon HS, in RSTS P2$_{EP300}$ hPSFs compared to HD. Arrowheads, nSBs; white rectangle, magnified cell containing nSBs. Quantification of the percentage of hPSFs containing nSBs ($n = 3$, 100–150 cells). Error bars, mean ± SEM; **$p = 0.0286$. Significance was calculated by two-sided Mann–Whitney test. Scale bar: 10 μM. Source data are provided as a Source Data file.

fibroblasts and the storage of the collection was approved and registered by the Département de la Recherche Clinique et du Développement (DRCD), Groupement Inter-régional de Recherche Clinique et d'Innovation d'Ile de France) (AP-HP), through consent for use for research (DC-2009-939).

The derivation of iPSCs from RSTS P1 and RSTS P2 Primary skin fibroblasts: the « Cellule de bioéthique, Direction générale de la recherche et de l'innovation » at the French Ministère de l'enseignement supérieur et de la Recherche (MESRI) delivered the CODECOH agreement (DC-2021-4446) that validated the derivation at the iPSC core facility of Nantes, and banking, storage and use of these iPSC lines at the Epigenetic and Cell Fate Center.

Purchased commercial healthy donor iPSCs IMR90-4 come from WiCell, USA; MTA 21-W0506 (female; fetal). The above-cited CODE-COH agreement DC-2021-4446) by the « Cellule de bioéthique" also approved the use of these commercial iPSCs form WiCell.

Human embryonic stem cell H1 and H9 anonymous cell lines are commercially available cell lines (WiCell; https://hpscreg.eu/cell-line/WAe001-A; https://hpscreg.eu/cell-line/WAe009-A). The use of Human ESC H9 (female) to generate cerebral organoids at the Lancaster Lab was approved by the UK Stem Cell Board. Human ESC lines (H1 male and H9 female cells) were used by the iPSC core facility of Nantes for producing RNA lysates that were used as positive controls for the characterization of iPSCs derived from RSTS patients P1 and P2. These hESC lines have been imported, banked and used under the agreement of Agence de la Biomédecine RE17_007.

For the derivation of the lymphoblastoid cell line coming from a healthy donor: the lymphoblastoid cell line LLD 138 is a kind gift from Prof. Evani-Viegas Pequignot (Institut Jacques Monod, Université Paris Diderot (now Université Paris Cité), a founder of our Epigenetics and Cell Fate Center. LLD 138 has been described in Almeida et al.[71]. At this time, patient's consent was not necessary to derive this cell line. For this reason, there is no information about the sex, nor the age of the donor.

For the derivation of lymphoblastoid cells from RSTS patient P3 (*EP300*) and RSTS patient P5 (*CREBBP*): these RSTS lymphoblastoid cell lines belong to a Biobank that was registered at the creation of the « Centre de Ressources Biologique"s CRB-BioJeL » and have the authorization to transfer material for scientific use, after approval by « Cellule de bioéthique, Direction générale de la recherche et de l'innovation » at the French Ministère de l'enseignement supérieur et de la Recherche (MESRI) delivered the CODECOH agreement (DC-2009-1044 and AC-2015-2579), for the generation, maintenance, and use of these cell lines for research without local ethical approval. The MTA for the use of these cell lines in the context of this study was given by CRB-BioJeL to Epigenetic and Cell Fate center (BB-0033-00016; May 05, 2018, Paris). For Patient RSTS P4 (*CREBBP*), the corresponding RSTS lymphoblastoid cell line, belongs to the « Génétique-Maladies Rares » (Genetics – Rare diseases) Biobank, which was registered at the creation of the « Bordeaux Centre de Ressources Biologique » (CRB), after its approval by the "Comité de Protection des Personnes du Sud-Ouest Outre-Mer III" (DC-2014-2164), for the generation, maintenance, and

use of these cell lines for research without local ethical approval. Cell access was provided by the Biobank, upon approval of the scientific project of this study.

See Fig. 6a for a description of the deletion or mutation carried by the patients.

PSFs from healthy donors HD1, HD2 (8-day-old males) were obtained from a collaborator. These primary skin fibroblast cells are described in Yehezkel et al.[72]. They come from anonymous gifts of foreskins, removed during circumcision, for which, at the time of the study by Yehezkel et al., no ethics committee approval was needed. No cell line was derived from these fibroblasts.

For mouse models, the project has been approved by the Animal Experimentation Ethical Committee Buffon (CEEA-40) and recorded under the following reference by the Ministère de l'Enseignement Supérieur, de la Recherche et de l'Innovation (#2016040414515579).

### Contact for reagent and resource sharing

More detailed information and requests for resources and reagents should be directed to and will be fulfilled by the co-corresponding authors: Aurélie de THONEL (aurelie.dethonel@univ-paris-diderot.fr), Lea SISTONEN (lea.sistonen@abo.fi), and Valérie MEZGER (valerie.mezger@univ-apris-diderot.fr).

See also Supplementary Methods for antibodies and plasmids and constructs

### Reagents and treatments

Proteasome inhibitor MG132 (Sigma-Aldrich; C2211ZLL) was used at a final concentration of 10 - 20 μM at the indicated times. HDACs inhibitor VPA (Interchim, AYJ060) was used at 1 mM for 3 h. The HAT inhibitor C646 (Sigma–Aldrich; SML0002) was used at a final concentration of 20 or 40 μM for 4 h. For all chemicals, DMSO was used as vehicle (control).

Heat-shock treatments were performed in water bath at 42 or 43 °C for the indicated times.

### Plasmid constructs

The human HSF2-Snap (WT/mutants) were constructed from the HSF2-Myc (WT/mutants) plasmid after digestion of the inserts by EcoRI and KpnI and cloning into the EcoRI and EcoRV sites in frame with the C-terminal Flag tag in pSNAPf plasmid using In-Fusion Kit (Clontech). The human HSF2-YFP was constructed by PCR and cloned into the XhoI and SalI sites in frame with the N-terminal YFP tag in EGFp-C1 plasmid using In-Fusion Kit (Clontech). All PCR-amplified products for both plasmids were sequenced to exclude the possibility of second site mutagenesis. The cDNA coding for the acetyltransferase domain of murine CBP (1097–1774) was a kind gift of Pr. Ricardo Dalla-Favera (Columbia University, New York) and was used to generate cDNA coding for key domains of CBP: Full-HAT (1096–1700), HAT (1322–1700), RING (1205–1279), PHD (1280–1321), Bromodomain (1096–1205), later subcloned in pet28a plasmid (Invitrogen) in order to produce 6 His-tagged proteins.

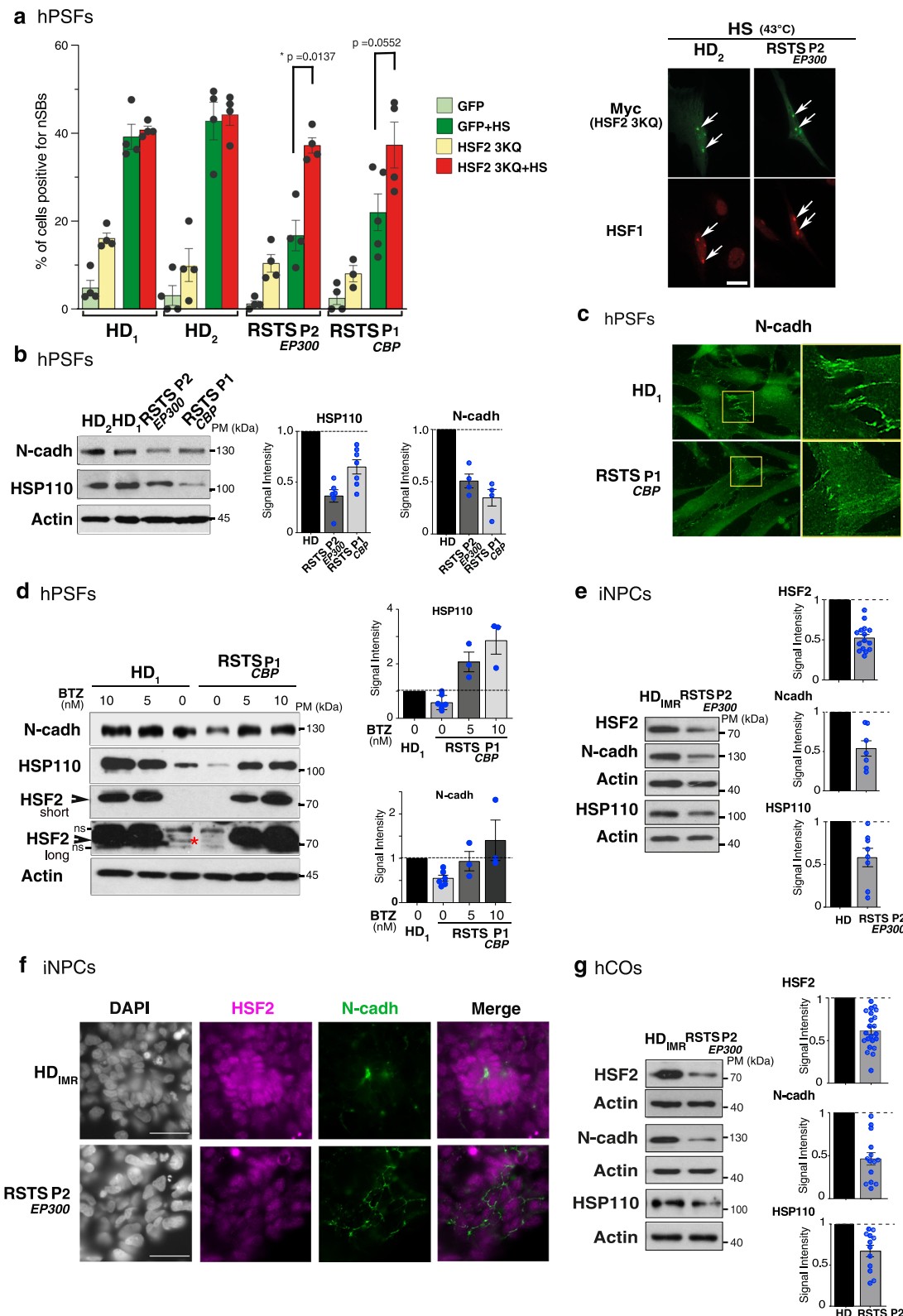

## Cell and cell line culture

Cell culture, transfections and treatments: murine Neuro2A (N2A, neuroblastoma, DSMZ # ACC 148), Hamster BHK (kindly provided by Dr. Leonhardt H and cultured as described in ref. 35, human HEK 293 T (ATCC®, CRL-11268™), U2OS (osteosarcoma, ATCC®, HTB-96™), U2OS-Crispr*HSF2*KO (2KO), SH-SY5Y (neuroblastoma, ATCC® CRL-2266™) were grown in DMEM (Lonza Group Ltd.) supplemented with 4,5 g/L

glucose and 10% fetal bovine serum (FBS, Life technology) in humidified atmosphere with 5% $CO_2$ at 37 °C. The generation and characterization of CRISPR/Cas9 *Hsf2*KO U2OS cells have been described in[23] (see also Supplementary Fig. 5c–e). hPSFs were grown in HAM's F10 supplemented with 12% FBS in humidified atmosphere with 5% $CO_2$ at 37 °C. Lymphoblastoid cells were grown in RPMI (Life technology) supplemented with 4.5 g/L glucose and 10% FBS with L-glutamine (Life

**Fig. 7 | HSF2-dependent dysregulated stress response and neurodevelopmental gene expression in cells from RSTS patients. a** Quantification of the percentage of cells containing nSBs (nuclear HSF1-positive dots) in HD₁, HD**₂**, RSTS P2_EP300_, and RSTS P1_CBP_ hPSFs transfected with HSF2 3KQ-Myc or GFP, and subjected or not to HS (1 h at 43 °C). HSF2 3KQ restores the induction of HS-induced nSBs in RSTS hPSFs. Transfection rate efficiencies: 16% for HD PSFs, 11% for RSTS hPSFs ($n = 4$, 100–200 cells per experiment). Error bars, mean ± SEM. *$p < 0.05$. Significance was calculated by two-sided multicomparison Friedmann Test. Representative immunofluorescence of RSTS hPSFs transfected with HSF2 3KQ-Myc upon HS showing nSBs identified with HSF1 (red) in transfected cells (Myc, green). Scale bar: 10 μm. See Supplementary Figs. 7 and 8a. **b** Representative immunoblots of protein extracts from HD₁, HD₂, RSTS P1_CBP_, and RSTS P2_EP300_ hPSFs showing reduced expression of N-cadherin and HSP110 levels in RSTS, compared to HD hPSFs. Quantification of N-cadherin and HSP110 levels in immunoblots, normalized to actin ($n = 3$). Error bars, mean ± s.d. **c** Representative Immunofluorescence of hPSFs

at cell–cell junctions (white dotted rectangles) showing that N-cadherin (green) is reduced in RSTS, compared to HD. Yellow rectangles, magnified areas. Scale bar: 20 μm. **d** Representative immunoblots of protein extracts from HD₂ and RSTS P1_CBP_ hPSFs treated by vehicle (0 nM) or BTZ (5 or 10 nM) for 22 h, showing increased HSF2 protein levels by subthreshold doses of BTZ, as well as restoration of HSP110 and N-cadherin levels in RSTS, compared to HD cells. *, endogenous HSF2 before treatment; short and long, different exposure times. Quantification of N-cadherin and HSP110 levels in immunoblots, normalized to actin ($n = 2$). Error bar, SEM. **e** Representative immunoblots of HD_IMR90_ and RSTS P2_EP300_ iNPCs, showing the reduction of levels of HSF2 and its targets in RSTS_EP300_, compared to HD iNPCs and hCOs. Quantification of the levels of HSF2, N-cadherin (N-cadh), and HSP110, detected in immunoblots, normalized to actin ($n = 3$). Error bars, mean ± SEM. **f** Representative immunofluorescence of HD_IMR90_ and RSTS P2_EP300_ iNPCs stained with N-cadherin (green) and HSF2 (purple). **g**. Same as in **e** but with D24(±1) hCOs. ($n = 3$). Error bars, mean ± SEM. Source data are provided as a Source Data file.

technology) in humidified atmosphere with 5% CO₂ at 37 °C and were kept at early passages to avoid putative compensation processes during ex vivo culture. See Fig. 6a for a description of the deletion or mutation carried by the lymphoblastoid cells. All cell lines were were tested to be mycoplasma free using Venor™ GeM Mycoplasma Detection Kit, PCR-based (Sigma–Aldrich).

iPSCs were grown in mTesR1 (Stem Cell Technologies) on plates coated with Matrigel (Corning). The differentiation of iPSCs to iNPCs was performed using the STEM diff™ SMADi neural induction kit and that of hCO using the STEM diff™ cerebral organoid kit, according to the manufacturer guidelines (Stem Cell Technologies).

### Mouse model

Specific pathogen-free C57BL/6N female mice were purchased from Janvier (Lyon, France) and maintained in sterile housing in accordance with the guidelines of the Ministère de la Recherche et de la Technologie (Paris, France). Rodent laboratory food and water were provided ad libitum. Experiments were performed in accordance with French and European guidelines for the care and use of laboratory animals. The invalidation strategy of the *Hsf2* gene has been described previously (ref. 12 *Hsf2^tm1Mmr* mouse strain in a C57BL/6 N background; here after *Hsf2^-/-*). *Hsf2* WT and *Hsf2^-/-* animals were produced by breeding *Hsf2* heterozygous mice.

### Immunoprecipitation and western blotting

Protein extracts from cells were prepared using a modified Laemmli buffer (5% sodium dodecyl sulfate, 10% glycerol, 32.9 mM Tris-HCl pH 6.8) supplemented with protease inhibitors (Sigma–Aldrich). Brain tissues were prepared with a lysis buffer (Hepes 10 mM pH 7.9; NaCl 0.4 M, EGTA 0.1 M; glycerol 5%, dithiothreitol [DTT] 1 mM, PMSF 1 mM, protease inhibitor [Sigma–Aldrich], phosphatase inhibitor [Roche]). Then, 30 μg of proteins from lysates were subjected to migration on 8–12% acrylamide gels and transferred on to polyvinylidene difluoride membranes (GE Healthcare Europe GmbH) in borate buffer (50 mM Tris-HCl and 50 mM borate) for 1 h 45 at constant voltage (48 V). The membranes were incubated with primary antibodies overnight at 4 °C, then washed in Tris-buffered saline–Tween 0.1% and incubated for 1 h with horseradish peroxidase (HRP)-coupled secondary antibody (Jackson Immunoresearch). The signal was revealed using a chemiluminescent reagent (Pierce® ECL Plus Western Blotting Substrate, Thermo Scientific) and was detected using hyperfilm (Hyperfilm™ ECL, Amersham Biosciences) and a film processor (Konica Minolta). Poly-ubiquitinated HSF2 was detected as described in ref. 21.

**For immunoprecipitation of exogenous proteins, using GFP/Myc-Trap.** GFP-Trap®-A (ChromoTek) contains a small recombinant fragment of alpaca anti-GFP-antibody, covalently coupled to the surface of agarose beads. It enables purification of any protein of interest

fused to GFP, eGFP, YFP, CFP or Venus. HEK 293 cells were transfected by a combination of YFP- or Myc-tagged hHSF2 and HA-tagged EP300, CBP (WT or DN) or GFP-tagged HDAC1, or mock vector, with XtremGENE HP Reagent (Sigma–Aldrich) following manufacturer's instructions. Cells were lysed in Lysis buffer (50 mM Hepes pH 8, 100 mM NaCl, 5 mM EDTA, Triton X-100 0.5%, Glycerol 10%, VPA (1 mM), DTT 1 mM, PMSF 1 mM, proteases inhibitors, phosphatase inhibitors [Roche]) and then, HSF2 was immunoprecipitated using anti-GFP- or anti-Myc-trap antibody, or as a control Trap®-A control (ChromoTek). Immunoprecipitated proteins were run on an 8% SDS-polyacrylamide gel, followed by an immunodetection of CBP or EP300 protein using anti-HA antibody. The amount of immunoprecipitated HSF2 was determined after reblot of the IP membrane with an anti-GFP or anti-Myc antibody. The amount of HSF2 and CBP or EP300 proteins, in the input samples, were detected with anti-GFP or Myc and anti-HA antibodies, respectively.

**For immunoprecipitation of endogenous proteins.** Brain cortices or organoids, or cells (N2a, SHSY-5Y) were lysed 30 min in Lysis buffer A (25 mM Hepes pH 8, 100 mM NaCl, 5 mM EDTA, Triton X-100 0.5%, 1 mM VPA, 1 mM PMSF, protease inhibitors, phosphatase inhibitors [Roche]). After centrifugation (15 min, 12 000 g) and preclearing, cell lysates were subjected to immunoprecipitation overnight using an anti-mouse HSF2 (Santa-Cruz) and a non-relevant IgG (Sigma–Aldrich) as a negative control that were pre-incubated 1 h at RT with Protein G UltraLink Resin beads (53132, Pierce). Protein complexes were then washed 4 times in wash buffer (25 mM Tris-HCl pH 7.5, 150 mM NaCl, 1 mM EDTA, Triton X-100 0.1% Glycerol 10%, 1 mM VPA, 1 mM PMSF, protease inhibitors, phosphatase inhibitors [Roche]), and suspended in 2× Laemmli buffer. After boiling, the immunoprecipitates were resolved in 8% SDS-PAGE and immunoblots were performed using an anti-rabbit pan acetyl-Lysine, anti-mouse HSF2 (Santa-Cruz), EP300 (Santa-Cruz) and CBP (CST). The amount of HSF2 and CBP or EP300 proteins in the input samples were detected with anti-mouse HSF2 and anti-rabbit CBP (CST) or EP300 (Santa-Cruz) antibodies.

### Biolayer interferometry

For in vitro protein-protein interaction experiments, we used biolayer interferometry technology (Octet Red, Forté-Bio, USA). Recombinant HSF2 (TP310751 Origen) was desalted (ZebaTM Spin Desalting Columns, 7 K molecular weight cutoff, 0.5 ml (1034–1164, Fisher Scientific, Germany)) and biotinylated at a molar ratio biotin/protein (3:1) for 30 min at room temperature (EZ-Link NHS-PEG4-Biotin [1189–1195, Fisher Scientific, Germany]). Excess Biotin was removed using ZebaTM Spin Desalting Columns. Biotinylated recombinant HSF2 was used as a ligand and immobilized at 100 nM on streptavidin biosensors after dilution in phosphate-buffered saline (PBS; 600 s). Interactions with desalted analytes diluted in PBS at 100 nM (recombinant CBP domains

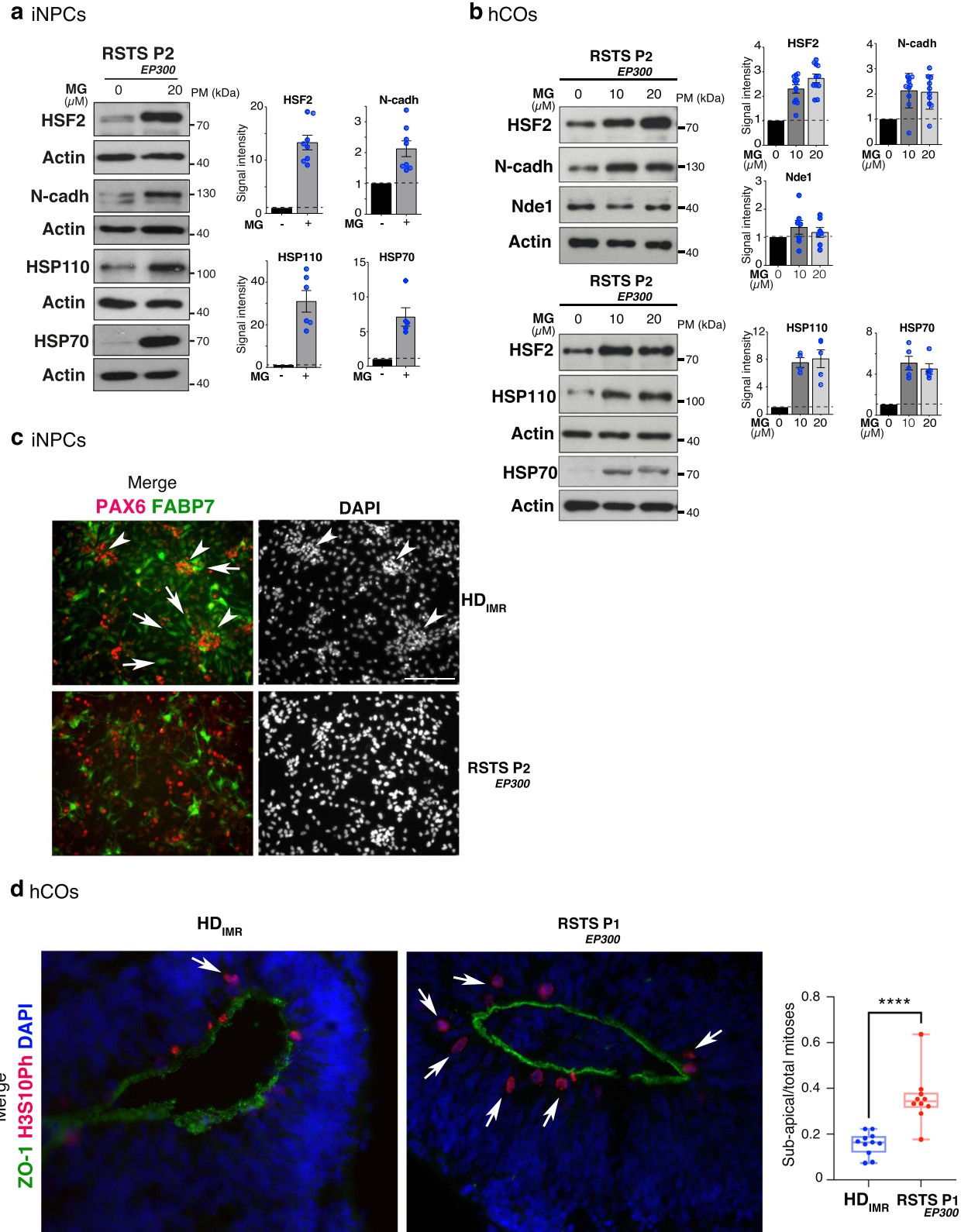

6 His-tag Full-HAT, Bromodomain [BD], RING, or HSP70 as a positive control [ADI-SPP-555, Enzo-Life Sciences]) were analyzed after association (600 s). All sensorgrams were corrected for baseline drift by subtracting a control sensor exposed to running buffer only. For Kd determination, each Kd was calculated with a 1:1 stoichiometry model using a global fit with Rmax unlinked by sensor (FortéBio, Data analysis software version 7.1.0.89).

**Immunohistochemistry and immunofluorescence**

hCO sections were prepared as described in Lancaster and Knoblich (2014)[30]. Mouse cortical tissues were fixed overnight at 4 °C in 4% paraformaldehyde (PFA). Cryoprotected and stored at −80 °C. 12 µm cryosections were stored at −80 °C. After antigen retrieval in citrate buffer (0.01 M citrate in 10% glycerol) 1 h at 65–70 °C, sections were washed, blocked in PBS with 1% horse serum, 0.1% Triton X-100 for 1 h

**Fig. 8 | Pharmacological augmentation of HSF2 levels restores its target expression in iNPCs and hCOs. a, b** Representative immunoblots of protein extracts from RSTS P2$_{EP300}$ iNPCs (**a**) or D25 hCOs (**b**) after treatment for 8 h (iNPCs) or 8h (hCOs) with vehicule (0) or MG132 (10 or 20 μM) showing the restoration of protein levels of HSF2 and its targets in the presence of MG132. Quantification of HSF2, N-cadherin, NDE1, HSP70, and HSP110 signal intensity in immunoblots, normalized to actin (*n* = 3). Error bars, mean ± SEM. **c** Representative immunofluorescence labeling of iNPCs by neural progenitor (PAX6) and radial glia (FABP7) markers (*n* = 3) showing the rosette-like, radial organization present in HD and lost in RSTS P2$_{EP300}$. Arrowheads, PAX6 positive groups of cells; arrows,

radially organized FABP7 positive cells. Scale bar: 50 μm. **d** Representative immunofluorescence of HD or RSTS P1$_{CBP}$ D25 hCOs stained for the VZ apical belt (ZO-1, green) and for mitotic progenitor cells (H3S10Ph, red). Arrows point to subapical mitoses. Scale bar: 50 μm. Quantification of the mean subapical mitoses, relative to total mitoses per hCO loop (H3S10Ph positive cells not in contact with the apical belt) (*n* = 3 independent hCO production runs; for HD, *n* = 11 hCOs, 52 loops, 526 mitoses; for RSTS, *n* = 8 hCOs, 43 loops, 275 mitoses). The box indicates the upper and lower quartiles and the whiskers indicate the 5th and 95th percentiles of the data. Error bars, mean ± s.d., *p* = 0.0001; ***p* < 0.0001. Significance was calculated by two-sided Mann–Whitney test. Source data are provided as a Source Data file.

at RT and primary antibodies were incubated ON at 4 °C. After washing, secondary antibodies coupled to fluorophore were incubated at RT for 1 h. When appropriate, a directly coupled primary antibody was then added after extensive washes for 1 h at RT or ON at 4 °C. Images were acquired by confocal microscopy LSM on a Leica SP5 system (IMAGOSEINE Imaging Platform at the Institut Jacques Monod) and were processed on FIJI. Average intensity projection of 3–4 z-slices (0.3 μm steps) are shown.

For iNPCs and hCOs, antigen retrieval was performed using citrate buffer (0.1 M sodium citrate pH 6.0, 10% glycerol, Tween 0.05%) for 1 h at 68 °C. Slices were washed, then saturated for 30 min for iNPCs and 1 h for hCOs with 1% horse serum in 0.1% PBS -Triton X-100 and incubated with primary antibody overnight at 4 °C. After washing in 0.1% PBS-Tween (iNPCs) or PBS-Triton X-100 (for hCOs), slices were incubated with corresponding secondary antibody and DAPI (120 ng/ml) for 1 h at room temperature. For data presented in Fig. 1, images were acquired by confocal microscopy as described above. For Figs. 7 and 8, iNPC or hCO images were taken by epifluorescence microsocopy on a Leica DMI 6000B (EPI² epifluorescence for epigenetics Platforme at UMR7216) and processed on ImageJ.

PSFs, in basal or heat-shock conditions, were fixed in 4% PFA on coverslip and stained with HSF1 (CST), HSF2 (Santa-Cruz), EP300 (Santa-Cruz), or N-cadherin (Proteintech) antibodies followed by a staining with the corresponding mouse or rabbit fluorescent secondary antibody (Jackson Immunoresearch). Images were acquired by epifluorescence microscopy and analyzed, as described above.

### Quantification of subapical mitosis

For each image considered, the loop or portion of loop to be analyzed was first identified in an unbiased way using DAPI and ZO-1 staining. We first delimited the apical and basal border of the area to be analyzed, then identified a subapical area, comprising two to three nuclei diameter from the apical border. Mitotic figures (PH3-positive cells) were scored as apical (touching the apical ZO-1 staining), subapical (within the subapical area), or basal (basal part of the loop). Mean subapical mitosis over total mitosis per hCO was then considered and analyzed. We analyzed hCOs from three independent experiments (see Fig. 8 and Supplementary Fig. 11 legends).

### Fluorescence three-hybrid assay

Fluorescence three-hybrid assay was performed according to[35]. BHK cells were transfected with constructs expressing expressing YFP-HSF2, CBP-HA, or EP300-HA, and GBP-LacI, using different combinations (ratio 1:1.5:2) at 70–80% confluency using reverse transfection by Lipofectamine 2000 (ThermoFisher Scientific), as indicated. Medium was changed after 4 h for all transfections. After 24 h, the cells were fixed in 4% PFA on coverslip and stained with mouse anti-HA (Covance) or rabbit anti-CBP antibody (Santa-Cruz), followed by a staining with mouse or rabbit fluorescent secondary antibody (Jackson Immunoresearch), respectively. Confocal microscopy images were taken on a confocal microscope Leica TCS SP5 (IMAGOSEINE Imaging Platform in Institut Jacques Monod) and images were analyzed using Fiji software (ImageJ2 v2;3;0/1.53k).

### RP-UFLC-based separation and quantification of CBP substrate peptides (HSF2) and their acetylated forms

For acetylation assays, we synthetized several 5-fluorescein amidite (5-FAM)-conjugated peptide substrates based on the human HSF2 sequence and containing various lysine residues of interest (Proteogenix):
- 5-FAM-SGIVK82QERD-NH2, referred to as K82 peptide
- 5-FAM-SSAQ135VQIR-NH2, referred to as K135 peptide
- 5-FAM-SLRRK197RPLL-NH2, referred to as K197 peptide

We also synthesized acetylated versions of these HSF2 peptides as standards. Samples containing HSF2 peptides and their acetylated forms were separated by RP-UFLC (Shimadzu) using Shim-pack XR-ODS column 2.0 ×100 mm 12 nm pores at 40 °C. The mobile phase used for the separation consisted of 2 solvents: A was water with 0.12% trifluoacetic acid (TFA) and B was acetonitrile with 0.12% TFA. Separation was performed by an isocratic flow depending on the peptide:
- 80% A/20% B, rate of 1 ml/min for K82 and K135
- 77% A/23% B, rate of 1 ml/min for K197

HSF2 peptide (substrate) and their acetylated forms (products) were monitored by fluorescence emission ($\lambda$ = 530 nm) after excitation at $\lambda$ = 485 nm and quantified by integration of the peak absorbance area, employing a calibration curve established with various known concentrations of peptides.

### In vitro acetyltransferase assay

To determine the activity of recombinant CBP-Full-HAT on HSF2 peptides, we used 96-wells ELISA plate (Thermofisher) and assays were performed in a total volume of 50 μL of acetyltransferase buffer (50 mM Tris pH 8, 50 mM NaCl) with 500 nM CBP-Full-HAT, 50 μM HSF2 peptides, and 1 mM DTT. Reaction was then started with the addition of 100 μM Acetyl-CoA (AcCoA) and the mixture was incubated 20 min at room temperature. Fifty microliters of HClO4 (15% in water, v/v) was used to stop the reaction and 10 μL of the mixture were injected into the RP-UFLC column for analysis. For time course studies, aliquots of the mother solution were collected at different time points and quenched with 50 μL of HClO4 prior to RP-UFLC analysis.

### Statistics

Data are displayed as means ± standard deviation (s.d.) or standard error of the mean (SEM). GraphPad Prism 8 (GraphPad Software, La Jolla, CA, USA) was used for statistical analyses. Statistical significance was assessed using the Mann–Whitney test for two groups (Fig. 5, Fig. 6, Supplementary Fig. 6; Fig. 8d, Supplementary Fig. 11h) or Friedmann Test (multicomparison, Fig.7). All statistical tests are two-sided. *p*-values below 0.05 are considered statistically significant.

"Supplementary Methods" are in the" Supplementary Information" file (see also the "Description of Additional Supplementary files").

### Reporting summary

Further information on research design is available in the Nature Portfolio Reporting Summary linked to this article.

## Data availability

All the data generated during this study are available from the corresponding authors on reasonable request. Source data are provided with this paper.

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

## Acknowledgements

We warmly thank the patients and their families for their participation in this study. We are grateful to H. LEONHARDT (Ludwig-Maximilians University, Munich, Germany) for F3H cellular and molecular tools, I. LEMASSON (East Carolina University, USA) for the KIX-GST constructs, and S. POLO (UMR7216) for SNAP-TAG vectors. We thank S. SELIG from the Technion (Israel) for primary skin fibroblasts from healthy donors. We thank the I. COUPRY and B. ARVEILER (CHU de Bordeaux, France) and Institut Médical Jérôme Lejeune for the gift of lymphoblastoid cells. We are grateful to A. DAVY and T. JUNGAS, for excellent training in human cerebral organoid production (UMR5547 Centre de Biologie Intégrative, CBI-CNRS, Toulouse, France). We thank the Platform Managers at the Epigenetics and Cell Fate Center for access to instruments and technical advice: L. FERRY (Functional Epigenomics (*EpiG* Platform), S. PIQUET (EPI$^2$ Microscopy Platform), and M. HENNION (Bioinformatics and Biostatistics), M. BOCEL and K. BOUHALI (enSCORE platform mutualized with and located at Institut Jacques Monod (IJM), Paris, France). We thank the Imagoseine (Imaging Platform) and Buffon animal housing facility at IJM. We than the Bioprofiler Platform (UMR8251 Biologie Fonctionnelle et Adaptative; in vitro acetylation assays). This study contributes to the Université de Paris IdEx #ANR-18-IDEX-0001 funded by the French Government through its "Investments for the Future" program and the Fédération Hospitalo-Universitaire "*Early Identification of Individual Trajectories in NeuroDevelopmental Disorders*" (FHU I2D2). (see Supplementary Acknowledgements). V.M. was funded by the CNRS (Projet International de Coopération Scientifique PICS 2013-2015), the Short Researcher Mobility France Embassy/MESRI-Finnish Society of Sciences and Letters, the Agence Nationale de la Recherche (« HSF-EPISAME », SAMENTA ANR-13-SAMA-0008-01), the Transversal Programs of Labex *Who am I?* AdT and VM were funded by the RUBINeuroStress ANR-19-CE16-0030 and Fondation de la recherche Médicale (FRM *Équipe labellisée* "Equ201903007924). V.M. was funded by the Labex *Who am I?* for a tranversal project on neural organoids. L.S. was

funded by the Academy of Finland, Sigrid Jusélius Foundation, Magnus Ehrnroth Foundation, Cancer Foundation Finland, and Åbo Akademi University. L.N. is Research Director of the F.R.S.-F.N.R.S. His work was supported by the F.R.S.-F.N.R.S. (Synet; EOS 0019118F-RG36), the Belgian Science Policy (IAP-VII network P7/20), and the ERANET Neuron STEM-MCD. R.A. was funded by Neuropôle Ile de France and Fondation ARC; F.M. by FRM and the CNRS, and ANR-13-SAMA-0008-01; K.D. and A.D. by the Ministère de l'Enseignement supérieur, de la Recherche et de l'Innovation (MESRI); A.D.T., D.S.D., and D.B. by CNRS PICS travel grant; G.P. and M.H. by ANR-13-SAMA-0008-01; C.C. by *RUBINeuroStress* ANR-19-CE16-0030; C.L. by FRM Equ201903007924; J.B. by the *Région Ile-de-France* (Cancéropôle IDF) and University Paris Cité; J.K.A. by Magnus Ehrnroth Foundation; M.C.P. by the Turku Doctoral Network in Molecular Biosciences and Magnus Ehrnroth Foundation. The supporting bodies played no role in any aspect of study design, analysis, interpretation, or decision to publish this data. C.C. was funded by a CIFRE grant from the Association Ksilink (Strasbourg, France), subsidized by the National Association of Research and Technology (ANRT - Association Nationale de la Recherche et de la Technologie, France) (grant CIFRE n° 2021/0560) and by RUBINeuroStress ANR-19-CE16-0030.

## Author contributions
A.d.T. was involved in the conceptual framework, contributed to the experimental design and performance, and data analyses of all experiments. V.D. designed and performed the confocal imaging of HSF2 and CBP/EP300 co-expression profile in the mouse developing cortex and hCOs, and contributed to the conceptual framework, experimental design and analysis of the phenotypic exploration of patient-derived hCOs, and reviewing of the manuscript. J.K.A., in collaboration with S.D.W., contributed to the conceptual and experimental strategy for and performed mass spectrometry identification of HSF2 acetylated residues. K.D. generated hCOs, designed and performed the biochemical and phenotypic characterization of hCOs and contributed to that of NPCs. J.B. and F.R.-L. designed and performed the analysis of in vitro acetylation of HSF2 peptides by RP-UFLC. C.C. and C.L. performed the biochemical and imaging characterization of patient-derived NPCs. M.S. contributed to the design of and G.P. performed SNAP-TAG experiments. J.K.A., A.L.A., and M.C.P. performed analysis of HSF2 mutant acetylation and ubiquitination. R.A., A.D., A.Vi., A.L.A., and F.M. contributed to initial experiments in HSF2 acetylation and interaction with CBP/EP300 in cellular and mouse model systems. L.C. and L.N. provided guidance to the V.M. laboratory for the reprogramming of RSTS*_EP300_* patient PSF-derived iPSCs and performed their characterization. I.L. and L.D. performed the reprogramming and characterization of RSTS*_CBP_* hPSFs. S.P., A.Ve., P.G., P.F., and D.L. diagnosed RSTS patient, performed skin biopsies or collected blood samples, and/or genotyped patient-derived primary or transformed cells. E.L., P.F., and D.L. generated patient-derived primary or transformed cells. M.C., under the supervision of J.G. and C.G., performed biolayer interferometry experiments. S.N. and C.B., under the co-supervision of A.dT., M.P., and O.T., performed in silico structural analyses of CBP/EP300 interaction with HSF2. M.L. provided paraffin-embedded and frozen hESC-derived hCOs, and RNA-Seq data on *HSF2* and *CREBBP/EP300* gene expression in hCOs. D.S.D. participated in initiating SNAP-TAG, genome-editing, and in silico investigations and co-supervised A.D. and F.M. V.M. and L.S. conceptualized and designed the study. V.M. wrote the paper, with the contribution of A.dT., V.D., and L.S.

## Competing interests
The authors declare no competing interests.

## Additional information

**Aurélie de Thonel** [1]✉, **Johanna K. Ahlskog** [2,3,20], **Kevin Daupin** [1,20], **Véronique Dubreuil** [1], **Jérémy Berthelet** [4], **Carole Chaput** [1,5], **Geoffrey Pires** [1], **Camille Leonetti** [1], **Ryma Abane** [1], **Lluís Cordón Barris** [6], **Isabelle Leray** [7], **Anna L. Aalto** [2,3], **Sarah Naceri** [1], **Marine Cordonnier** [8,9,10], **Carène Benasolo** [1], **Matthieu Sanial** [11], **Agathe Duchateau** [1], **Anniina Vihervaara** [2,3,18], **Mikael C. Puustinen** [2,3], **Federico Miozzo** [1,19], **Patricia Fergelot** [12], **Élise Lebigot** [13], **Alain Verloes** [14,15], **Pierre Gressens** [14], **Didier Lacombe** [12], **Jessica Gobbo** [8,9,10], **Carmen Garrido** [8,9,10], **Sandy D. Westerheide** [16], **Laurent David** [7], **Michel Petitjean** [4], **Olivier Taboureau** [4], **Fernando Rodrigues-Lima** [4], **Sandrine Passemard** [14], **Délara Sabéran-Djoneidi** [1], **Laurent Nguyen** [6], **Madeline Lancaster** [17], **Lea Sistonen** [2,3,21] & **Valérie Mezger** [1,21]✉

[1]Université de Paris, CNRS, Epigenetics and Cell Fate, F-75013 Paris, France. [2]Faculty of Science and Engineering, Cell Biology, Åbo Akademi University, Turku, Finland. [3]Turku Bioscience Centre, University of Turku and Åbo Akademi University, Turku, Finland. [4]Université de Paris, CNRS, Unité de Biologie

Fonctionnelle et Adaptative, Paris, France. [5]Ksilink, Strasbourg, France. [6]Laboratory of Molecular Regulation of Neurogenesis, GIGA-Stem Cells and GIGA-Neurosciences, Interdisciplinary Cluster for Applied Genoproteomics (GIGA-R), University of Liège, CHU Sart Tilman, Liège, Belgium. [7]Université de Nantes, CHU Nantes, Inserm, CNRS, SFR Santé, Inserm UMS 016, CNRS UMS 3556, F-44000 Nantes, France. [8]INSERM, UMR1231, Laboratoire d'Excellence LipSTIC, Dijon, France. [9]University of Bourgogne Franche-Comté, Dijon, France. [10]Département d'Oncologie médicale, Centre Georges-François Leclerc, Dijon, France. [11]CNRS, UMR 7592 Institut Jacques Monod, F-75205 Paris, France. [12]Department of Medical Genetics, University Hospital of Bordeaux, Bordeaux, France and INSERM U1211, University of Bordeaux, Bordeaux, France. [13]Service de Biochimie-pharmaco-toxicologie, Hôpital Bicêtre, Hopitaux Universitaires Paris-Sud, 94270 Le Kremlin Bicêtre, Paris-Sud, France. [14]Université de Paris, INSERM, NeuroDiderot, Robert-Debré Hospital, F-75019 Paris, France. [15]Genetics Department, AP-HP, Robert-Debré University Hospital, Paris, France. [16]Department of Cell Biology, Microbiology, and Molecular Biology, College of Arts and Sciences, University of South Florida, Tampa, FL, USA. [17]MRC Laboratory of Molecular Biology, Cambridge Biomedical, Campus, Cambridge, UK. [18]Present address: KTH Royal Institute of Technology, Stockholm, Sweden. [19]Present address: Neuroscience Institute-CNR (IN-CNR), Milan, Italy. [20]These authors contributed equally: Johanna K. Ahlskog, Kevin Daupin. [21]These authors jointly supervised this work: Lea Sistonen, Valérie Mezger.
✉e-mail: aurelie.dethonel@univ-paris-diderot.fr; valerie.mezger@univ-paris-diderot.fr

