## [Peer Review File · Nature Communications]

CBP-HSF2 structural and functional interplay in Rubinstein-Taybi neurodevelopmental disorderREVIEWER COMMENTS

Reviewer #1 (Remarks to the Author):

This manuscript represents a major advance in understanding how mutations in a global protein acetylase, underlying a rare neurodevelopmental disorder, impacts cellular stress protection and neurodevelopment. Launching from previous work by these labs on the role and regulation of the lesser appreciated HSF, HSF2, in neuronal development, stress protection and beyond, this report provides an excellent blend of genetics, biochemistry and cell biology approaches to present rigorous, compelling data demonstrating a role for HSF2 in RTS. The authors present a very strong case for specific interactions between HSF2 within the KIX domain found in the coiled-coil region and CBP/p300 and validate the interaction with biochemical evidence, molecular modeling and mutagenesis. Moreover, while previous reports demonstrated the regulation of HSF2 via protein degradation, this work deciphers key mechanisms that regulate this process via protein lysine acetylation. Finally, the current work underscores the contributions of HSF2, and its dysregulation in RSTS, to stress-regulated gene expression and its impact in neuronal cell function. Overall, the results reported here are novel and highly impactful to the fields of stress regulation, neurodevelopmental processes and rare diseases.

Reviewer #2 (Remarks to the Author):

It is very important to elucidate the molecular mechanism of Rubinstein-Taybi neurodevelopment disorder, in particular how specific mutations in CBP/EP300 affect downstream targets, which in turn affect neurodevelopmental processes. In such a study, both biochemical/cellular dissection of molecular cascades and the validation of the biological relevance of the elucidated molecular cascades (in particular, the latter) are crucial.

In Figure 1, the authors showed a similar expression pattern of CBP/EP300 and HSF2. It is unclear why the authors focused on HSF2 specifically. Furthermore, a "similar" expression pattern may be seen in many proteins with CBP/EP300. In neuroscience, cell-type specificity is very important. However, the resolution of histology (cell-type specificity) might not be well addressed in the present study, and the biochemical co-IP cannot address this question.

Biochemical characterization for specific acetylation of HSF2 protein is good. This may be affected by CBP/EP300 at least in cell cultures. However, it is unclear if this happens in neurodevelopment context in vivo.

The most unfortunate point in the manuscript is that biological and functional impact of the molecular cascade (CBP/EP300 to HSF2) in neurodevelopment is not addressed. The authors are recommended to use in utero gene transfer to knockdown the target protein to show the impact of the target molecule(s). This can be combined with rescue test by co-introducing either wild-type or mutant (acetylation-dead etc) construct. The authors do not need to stick to this methodology, but may use organoids or other neurodevelopment-relevant biological systems for functional assays. However, without including the data of

such functional validation in a neurodevelopmental context, it is very difficult to make readers in neuroscience feel satisfied in a general journal such as NC. Otherwise, this manuscript (although pretty limited in the scope, well done in biochemistry and limited cell biology) may appeal to readers in biochemical and cellular biology journals.

Reviewer #3 (Remarks to the Author):

The authors presented data showing altered acetylation of HSF2 possibly instigates the neurodevelopmental deficits in Rubinstein-Taybi Syndrome (RSTS). They found that mutations of CBP or EP300 impair the stability of the short-lived HSF2 protein and in turn decreases the expression of genes that affect neural development. They also employed BTZ to upregulate HSF2 levels and restore the expression of HSP110 and N-Cadherin for developing treatment of RSTS. Their work reveals additional regulatory machinery of neural development and its pathological relevance to RSTS. The strength of this work lies on their elaborative demonstration on the regulatory relationship between CBP and HSF. However, it also derails from further clarification of how the dynamic regulatory pathway causes the variable phenotypes of RSTS.

Several aspects need further clarification.

1. As HSF2 pleiotropically regulates neural development, such as controlling number of cortical radial glial cells and radial neuronal migration (lines 6-8 on page 5). It would be valuable if the authors present some data on this. Are there any relevant alterations in RSTS patients, animal models or brain organoids? Such kind of assays would be helpful to elucidate the machinery of RSTS and verify the effectiveness of the proposed treatment development.
2. The authors show in RSTS cells that BTZ can increase the amount of N-Cadherin. They then propose such a drug can be further developed as a potential therapeutic drug. Additional assays, such as on mouse model of CBP/EP300 mutation or organoids of RSTS, plus in depth analysis are needed. It would be important to see in mouse model whether this drug can improve the cognitive function?
3. Some details need further clarity. The authors mentioned that HSF2 is acetylated by CBP/EP300 at three main lysine residues (K128, K135 and K197). In Fig. 2 and Fig. S2, only data of synthetic HSF2 peptide containing K82, K135 and K197 residues are shown, and no data about K128 residue is presented.
4. In Fig.1 and Fig S1, although they show the general staining sites in organoid slices, there is no direct proof of the colocalization of CBP /HSF2 and markers of neural stem cells or neuronal cells.
5. The last three lines of P7, "impact of in silico mutations of the K177, K180, F181, V183 residues", K180 should be Q180.

Reviewer #4 (Remarks to the Author):

HATs and HDACs conduct protein acetylation, including both histones and other proteins. It is believed that HATs and HDACs are important in brain development and also in neuron functionality. HDACs inhibitors have been long used in clinics to treat various neurological deficits including memory formation, mood, drug addiction, and neuroprotection, suggesting a role for HAT/HDAC in neuron homeostasis. In addition, mutations of the HAT complexes have been linked to human neurological deficits as well as to neurodevelopmental disorders. Rubinsten-Taybi Syndrome (RSTS) has been shown to be associated with mutations in HAT CBP/P300. However, how the epigenetic modulation, ie., by CBP/P300, regulates brain development, remains largely unknown.

The authors devised molecular and biochemical experiments to decipher the molecular mechanism by which CBP/P300 regulates its downstream pathways. Intriguingly, the authors discovered that CBP/P300 directly interact and acetylate HSF2 (heat shock transcription factor), thereby its stability in response to heat shock or stress. The authors conducted comprehensive experiments, ranging from human brain organoid, mapping of protein-protein interactions and acetylation sites. They also used human (control donor and RSTS patients) materials and clinically approved drugs to verify the mechanisms that they discovered. The conclusion is supported by the data. The study presents a significant advance of the concept. However, some points should be addressed.

Specific comments

1. In order to view the colocalization of HSF2, CBP/P300 merge of different channels or stainings are needed. Fig 1B-C. The neuronal markers (eg. Tuj1) and NPC markers (eg. SOX2) should be provided in the same figure panels. Fig S1B-C. Merge images are needed.
2. Fig 1E-F. Many IP experiments showed separate Input blots. However, it is difficult to judge how good or abundant the interaction is. How many repeat experiments were done for Fig 1F?
3. Fig 3D-E. How many cells scored for the percentages presented?
4. Fig 5E. What are these levels (WT and 3KR) compared with 3KQ? P values need to be calculated and need more repeats (N=2 at the moment).
5. Fig 6. It is clear that HSP70 is low with or without acute HS, compare to HD samples. However, after recovery, HSP70 was greatly increased and even with high magnitudes. It is also clear that HS did not induce high HSP70 in HD (CTR vs HS). I wonder how the authors quantified these signals (right panel). Is something wrong here? It was not correctly stated in the main text (page 9, para. 2).
6. Fig 7. (A) The transfection efficiency and ectopic expression level of 3KQ should be controlled and documented. P values need to be calculated after total N number increased (currently N=2). (D, F) A quantification is needed, in order to make a firm statement.
7. Fig S7. Zoom in images would help to validate the statement.

REVIEWER COMMENTS

Reviewer #1 (Remarks to the Author):

This manuscript represents a major advance in understanding how mutations in a global protein acetylase, underlying a rare neurodevelopmental disorder, impacts cellular stress protection and neurodevelopment. Launching from previous work by these labs on the role and regulation of the lesser appreciated HSF, HSF2, in neuronal development, stress protection and beyond, this report provides an excellent blend of genetics, biochemistry and cell biology approaches to present rigorous, compelling data demonstrating a role for HSF2 in RTS. The authors present a very strong case for specific interactions between HSF2 within the KIX domain found in the coiled-coil region and CBP/p300 and validate the interaction with biochemical evidence, molecular modeling and mutagenesis. Moreover, while previous reports demonstrated the regulation of HSF2 via protein degradation, this work deciphers key mechanisms that regulate this process via protein lysine acetylation. Finally, the current work underscores the contributions of HSF2, and its dysregulation in RSTS, to stress-regulated gene expression and its impact in neuronal cell function. Overall, the results reported here are novel and highly impactful to the fields of stress regulation, neurodevelopmental processes and rare diseases.

Reviewer #2 (Remarks to the Author):

It is very important to elucidate the molecular mechanism of Rubinstein-Taybi neurodevelopment disorder, in particular how specific mutations in CBP/EP300 affect downstream targets, which in turn affect neurodevelopmental processes. In such a study, both biochemical/cellular dissection of molecular cascades and the validation of the biological relevance of the elucidated molecular cascades (in particular, the latter) are crucial.

In Figure 1, the authors showed a similar expression pattern of CBP/EP300 and HSF2. It is unclear why the authors focused on HSF2 specifically. Furthermore, a "similar" expression pattern may be seen in many proteins with CBP/EP300. In neuroscience, cell-type specificity is very important. However, the resolution of histology (cell-type specificity) might not be well addressed in the present study, and the biochemical co-IP cannot address this question.

Biochemical characterization for specific acetylation of HSF2 protein is good. This may be affected by CBP/EP300 at least in cell cultures. However, it is unclear if this happens in neurodevelopment context *in vivo*.

The most unfortunate point in the manuscript is that biological and functional impact of the molecular cascade (CBP/EP300 to HSF2) in neurodevelopment is not addressed. The authors are recommended to use *in utero* gene transfer to knockdown the target protein to show the impact of the target molecule(s). This can be combined with rescue test by co-introducing either wild-type or mutant (acetylation-dead etc) construct. The authors do not need to stick to this methodology, but may use organoids or other neurodevelopment-relevant biological systems for functional assays. However, without including the data of such functional validation in a neurodevelopmental context, it is very difficult to make readers in neuroscience feel satisfied in a general journal such as NC. Otherwise, this manuscript (although pretty limited in the scope, well done in biochemistry and limited cell biology) may appeal to readers in biochemical and cellular biology journals.

Reviewer #3 (Remarks to the Author):

The authors presented data showing altered acetylation of HSF2 possibly instigates the neurodevelopmental deficits in Rubinstein-Taybi Syndrome (RSTS). They found that mutations of CBP or EP300 impair the stability of the short-lived HSF2 protein and in turn decreases the expression of genes that affect neural development. They also employed BTZ to upregulate HSF2 levels and restore the expression of HSP110 and N-Cadherin for developing treatment of RSTS. Their work reveals additional regulatory machinery of neural development and its pathological relevance to RSTS. The strength of this work lies on their elaborative demonstration on the regulatory relationship between CBP and HSF. However, it also derails from further clarification of how the dynamic regulatory pathway causes the variable phenotypes of RSTS.

Several aspects need further clarification.

1. As HSF2 pleiotropically regulates neural development, such as controlling number of cortical radial glial cells and radial neuronal migration (lines 6-8 on page 5) . It would be valuable if the authors present some data on this. Are there any relevant alterations in RSTS patients, animal models or brain organoids? Such kind of assays would be helpful to elucidate the machinery of RSTS and verify the effectiveness of the proposed treatment development.
2. The authors show in RSTS cells that BTZ can increase the amount of N-Cadherin. They then propose such a drug can be further developed as a potential therapeutic drug. Additional assays, such as on mouse model of CBP/EP300 mutation or organoids of RSTS, plus in depth analysis are needed. It would be important to see in mouse model whether this drug can improve the cognitive function?
3. Some details need further clarity. The authors mentioned that HSF2 is acetylated by CBP/EP300 at three main lysine residues (K128, K135 and K197). In Fig. 2 and Fig. S2, only data of synthetic HSF2 peptide containing K82, K135 and K197 residues are shown, and no data about K128 residue is presented.
4. In Fig.1 and Fig S1, although they show the general staining sites in organoid slices, there is no direct proof of the colocalization of CBP /HSF2 and markers of neural stem cells or neuronal cells.
5. The last three lines of P7, “impact of in silico mutations of the K177, K180, F181, V183 residues”, K180 should be Q180.

Reviewer #4 (Remarks to the Author):

HATs and HDACs conduct protein acetylation, including both histones and other proteins. It is believed that HATs and HDACs are important in brain development and also in neuron functionality. HDACs inhibitors have been long used in clinics to treat various neurological deficits including memory formation, mood, drug addiction, and neuroprotection, suggesting a role for HAT/HDAC in neuron homeostasis. In addition, mutations of the HAT complexes have been linked to human neurological deficits as well as to neurodevelopmental disorders.

Rubinsten-Taybi Syndrome (RSTS) has been shown to be associated with mutations in HAT CBP/P300. However, how the epigenetic modulation, ie., by CBP/P300, regulates brain development, remains largely unknown.

The authors devised molecular and biochemical experiments to decipher the molecular mechanism by which CBP/P300 regulates its downstream pathways. Intriguingly, the authors discovered that CBP/P300 directly interact and acetylate HSF2 (heat shock transcription factor), thereby its stability in response to heat shock or stress. The authors conducted comprehensive experiments, ranging from human brain organoid, mapping of protein-protein interactions and acetylation sites. They also used human (control donor and RSTS patients) materials and clinically approved drugs to verify the mechanisms that they discovered. The conclusion is supported by the data. The study presents a significant advance of the concept. However, some points should be addressed.

Specific comments

1. In order to view the colocalization of HSF2, CBP/P300 merge of different channels or stainings are needed. Fig 1B-C. The neuronal markers (eg. Tuj1) and NPC markers (eg. SOX2) should be provided in the same figure panels. Fig S1B-C. Merge images are needed.
2. Fig 1E-F. Many IP experiments showed separate Input blots. However, it is difficult to judge how good or abundant the interaction is. How many repeat experiments were done for Fig 1F?
3. Fig 3D-E. How many cells scored for the percentages presented?
4. Fig 5E. What are these levels (WT and 3KR) compared with 3KQ? P values need to be calculated and need more repeats (N=2 at the moment).
5. Fig 6. It is clear that HSP70 is low with or without acute HS, compare to HD samples. However, after recovery, HSP70 was greatly increased and even with high magnitudes. It is also clear that HS did not induce high HSP70 in HD (CTR vs HS). I wonder how the authors quantified these signals (right panel). Is something wrong here? It was not correctly stated in the main text (page 9, para. 2).
6. Fig 7. (A) The transfection efficiency and ectopic expression level of 3KQ should be controlled and documented. P values need to be calculated after total N number increased (currently N=2). (D, F) A quantification is needed, in order to make a firm statement.
7. Fig S7. Zoom in images would help to validate the statement.

RESPONSE TO REVIEWERS

REVIEWER 1 (REMARKS TO THE AUTHOR):

This manuscript represents a major advance in understanding how mutations in a global protein acetylase, underlying a rare neurodevelopmental disorder, impacts cellular stress protection and neurodevelopment. Launching from previous work by these labs on the role and regulation of the lesser appreciated HSF, HSF2, in neuronal development, stress protection and beyond, this report provides an excellent blend of genetics, biochemistry and cell biology approaches to present rigorous, compelling data demonstrating a role for HSF2 in RTS. The authors present a very strong case for specific interactions between HSF2 within the KIX domain found in the coiled-coil region and CBP/p300 and validate the interaction with biochemical evidence, molecular modeling and mutagenesis. Moreover, while previous reports demonstrated the regulation of HSF2 via protein degradation, this work deciphers key mechanisms that regulate this process via protein lysine acetylation. Finally, the current work underscores the contributions of HSF2, and its dysregulation in RSTS, to stress-regulated gene expression and its impact in neuronal cell function. Overall, the results reported here are novel and highly impactful to the fields of stress regulation, neurodevelopmental processes and rare diseases.

We thank the Reviewer for its very positive statement of our work and of its impact, which has been a great encouragement for us.

REVIEWER 2

It is very important to elucidate the molecular mechanism of Rubinstein-Taybi neurodevelopment disorder, in particular how specific mutations in CBP/EP300 affect downstream targets, which in turn affect neurodevelopmental processes. In such a study, both biochemical/cellular dissection of molecular cascades and the validation of the biological relevance of the elucidated molecular cascades (in particular, the latter) are crucial.

Point I. In **Figure 1**, the authors showed a similar expression pattern of CBP/EP300 and HSF2. It is unclear why the authors focused on HSF2 specifically. Furthermore, a "similar" expression pattern may be seen in many proteins with CBP/EP300. In neuroscience, cell-type specificity is very important. However, the resolution of histology (cell-type specificity) might not be well addressed in the present study, and the biochemical co-IP cannot address this question.

Figure 1A. We understand the Reviewer's concern and thank her/him for her/his advice.

1) Global Expression levels of HSF2 in human cerebral organoids

We focused on HSF2, because, HSF2 is involved in normal (unstressed) prenatal cortical development, as are CBP and p300 (Chan and La Thangue 2001; Lopez-Atalaya et al., 2014; Wang et al., 2010). In support of this choice, CBP or EP300 mutations, on the one hand, and HSF2 mutations on the other hand, have been reported to cause neurodevelopmental syndrome (RSTS and Angelman Syndrome; Lopez-Atalaya et al., 2014 and Aguilera et al. 2021). In contrast, HSF1 is not involved in normal prenatal cortical development: this is the reason why we have focused on HSF2 in this study. Chang et al., 2006; El Fatimy et al., 2014; reviewed in Duchateau et al., 2020; please, see introduction, second paragraph). In contrast to HSF2, HSF1 is present, but not active for DNA-binding in the normal cortex. HSF1 only acquires DNA-binding activity and transcriptional activity in response to prenatal stress (Hashimoto-Torii et al., 2014; El Fatimy et al., 2014; reviewed in Duchateau et al., 2020). To avoid any confusion at this step, we have:

- added a sentence in the Introduction (page 6, first paragraph /lanes 210-212):

While no specific function has been attributed to HSF1 during the prenatal development, in physiological conditions, it is involved in spinogenesis and neurogenesis during the postnatal development of the murine hippocampus (Uchida et al., 2011).

- in the "Results" section, we have now started this paragraph (lanes 238-260) by describing new data on the analysis of HSF2 and CBP/EP300 colocalization in neural cells in human cerebral organoids (hCOs; revised Fig. 1A and moved the WB analysis of the previous Fig. 1A, showing HSF2 and HSF1 expression profile to revised Fig. 1B; please, see also below, point 3).

2) CBP/EP300 and HSF2 and cell-specific expression patterns. (Please, see also our answer to Reviewer #4, point 1)

To strengthen our data according to the Reviewer's suggestion and to identify in which neural cell types this colocalization might occur, we have performed confocal analyses in hCOs of the co-labelling of HSF2 and CBP/EP300 with markers: SOX2 and PAX6 for neural progenitor cells; TBR1 and Beta III-tubulin for young neurons.

First, we show that HSF2 is present in nuclei of the hCO proliferative (germinal) layer (PL, labelled by PAX6 or SOX2) and neuronal layers (NL; labelled by TBR1 or beta III-TUBULIN) and that CBP/EP300 are also observed in nuclei of these two compartments (revised Fig. 1A). We also confirmed that CBP localized in both in the PL and NL zones, using a CBP-specific antibody*. In addition, our data indicate that many of these nuclei are co-stained by HSF2 and CBP/EP300 (revised Fig. 1A). We therefore think that we have addressed the question of cell-specificity for the expression of these two proteins, which coexist in nuclei both in the hCO PL and NL zones.

In addition, we have also addressed the co-localization of HSF2 and CBP/EP300 and its cell-specificity in the developing mouse cortex at E11 and E15. Importantly, using confocal analyses, we found similar cell-specificity in co-staining for HSF2 with CBP/p300, which reinforces our data in hCOs (revised Supplementary Fig. 1A).

Notably, we do not address the expression profile of these proteins neither at late time-points of cortical development, stages at which we know that HSF2 levels are much decreased, nor in the adult cortex where the HSF1 is the sole active HSF, in contrast to the prenatal period (Duchateau et al., 2020).

*NB: Given the nature of commercial antibodies, we couldn't perform co-labelling of HSF2 and CBP in the hCOs, therefore we used an antibody that recognized both CBP and EP300.

3) To date, there were no data in the literature about the expression of neither the HSF2 protein, nor HSF1 in human cerebral organoids (hCOs). We thus thought that showing HSF1 and HSF2 global patterns of expression would be informative to revised Fig. 1B; (previously in Fig. 1A; see also point 1). We moved the RNA-seq data, formerly presented in Supplementary Fig. 1A, to revised Supplementary Fig. 1B, so that confocal analyses showing colocalization of HSF2 and CBP/EP300 in the mouse cortex would come first, as they are more informative for the purpose of the study, dealing with the interaction between HSF2 and CBP/EP300.

Altogether, we thus modified the result section accordingly (page 7, lanes 238-260).

Point II. The most unfortunate point in the manuscript is that biological and functional impact of the molecular cascade (CBP/EP300 to HSF2) in neurodevelopment is not addressed. The authors are recommended to use in utero gene transfer to knockdown the target protein to show the impact of the target molecule(s). This can be combined with rescue test by co-introducing either wild-type or mutant (acetylation-dead etc.) construct. The authors do not need to stick to this methodology, but may use organoids or other neurodevelopment-relevant biological systems for functional assays. However, without including the data of such functional validation in a neurodevelopmental context, it is very difficult to make readers in neuroscience feel satisfied in a general journal such as NC. Otherwise, this manuscript (although pretty limited in the scope, well done in biochemistry and limited cell biology) may appeal to readers in biochemical and cellular biology journals.

We agree with Reviewer #2 that addressing the biological and functional impact of the CBP/EP300-to-HSF2 cascade would of course improve the manuscript for an audience of neuroscientists.

We have seriously considered *in utero* electroporation, but the constant threats of shut-down of activities in animal houses have orientated us to a more amenable approach that could be set in the lab, during the pandemic period. As kindly suggested by the reviewer, we thus have developed 3D hCO models from iPSCs, derived from either healthy donors (HD) or RSTS-patients, as well as 2D culture of iPSC-derived NPCs (iNPCs). For this we derived high-quality iPSC clones RSTS_{CBP} [P1] and RSTS_{EP300} [P2] patient primary skin fibroblasts (hPSFs), and used IMR90-4 iPSCs as HD (Benito-Kwiecinski et al., 2021; revised Supplementary Fig. 9; Nantes iPSC Platform, France), whose pluripotency properties, genomic and chromosomal integrity were carefully checked at different banking steps (Supplementary Fig. 9). Their ability to generate iNPCs and hCO using *Stemcell Technologies* workflows, based on the protocol by Lancaster et al. (2014) were also assessed (Supplementary Fig. 10A-C). Please, see pages 11-13 (lanes 476-560), in the manuscript.

We recapitulated the features of the CBP/EP300-to-HSF2 cascade found in hPSFs (previous Figure 7), in both iNPC and hCO neural systems. The new data are described in revised Fig. 7E-G, Fig. 8, and Supplementary Fig. 10D,E and Supplementary Fig. 11) and in the corresponding (highlighted) "Results", "Legends" and "Discussion" sections in the manuscript. To summarize findings:

1) We demonstrate that, as it is the case for RSTS primary skin fibroblasts (hPSFs), the levels of HSF2 protein are lower in RSTS iNPC 2D-cultures, as are those of its targets compared with the HD counterparts, both in WB and IF experiments (revised Fig. 7E,F and Supplementary Fig. 10D). We also demonstrate that RSTS hCOs exhibit lower levels of HSF2, compared to HD hCOs (revised Fig. 7G and Supplementary Fig. 10E). Please, see page 12, lanes 500-504.

2) We demonstrate that the reduction of HSF2 in RSTS iNPC 2D cultures and in hCOs is correlated with diminished HSP110 and N-cadherin levels (revised Fig. 7E-G and Supplementary Fig. 10D,E). We also observed differences in the distribution of N-cadherin in RSTS iNPCs, compared to HD counterparts (Fig. 7F), but analysis redistribution of N-cadherin would clearly deserve deeper analyses that represent an entire study by itself about the role of HSF2 on N-Cadherin and its other targets, which is currently the focus of another project. Please, see page 12, lanes 500-504.

3) We also “rescued” (increased) HSF2 levels in RSTS iNPCs and hCOs.

a) We tried to rescue RSTS iNPCs by transiently transfecting them, using the HSF2 3KQ acetylated mimic. Although the empty control vector was nicely transfected as assessed by GFP-stained iNPCs, the “rescue” construct was not, despite the fact that we used different plasmid preparation and performed different transfection assays. We think that this might be due to the fact that HSF2 levels are tightly regulated in the neural context (at the transcriptional, posttranscriptional and post-translational levels, please see introduction) and that increasing HSF2 levels in a rather uncontrolled manner, as it is the case in transient transfection experiments, might be deleterious for iNPCs. For these reasons, we believe that such approach requires a CRISPR/Cas9 KI- strategy, in order to ensure a more physiological control of HSF2 levels in these neural cells, whose practical implementation is under progress, but beyond the timing of a revision process. Indeed, it implies to mute the 3 major K residues into Q, two of which are located at a distance of around 5kb, at the *HSF2* gene locus.

b) Thus, to circumvent this, we restored HSF2 levels in RSTS, pharmacologically.

Notably, we focused on the use of MG132, because the use of BTZ could not be implemented in these neurodevelopmental models: indeed, BTZ (or its chemically very close derivative, marizomib) unexpectedly failed to induce HSF2 protein levels, in iNPCs and hCOs, at different stages. One possibility is that BTZ has such a strong effect on HSF2 levels (Rossi et al., 2014; Joutsen et al., 2020) that compensatory mechanisms might prevent the marked increase in HSF2 levels in neurodevelopmental contexts very sensitive to HSF2 levels, in contrast to what is classically observed in non-neural cell systems (Rossi et al., 2014; Joutsen et al., 2020). In addition, BTZ and MG132 can differently operate through their mechanisms of inhibition of proteasomal activity and the reversibility of their action (Harbouri et al., 2017). Please, see page 12, lanes 504-532. We added a paragraph in the discussion, page 16, lanes 677-684).

To pharmacologically increase HSF2 levels, we thus treated iNPCs 2D-cultures or hCOs with the proteasome inhibitor MG132 for 8 hours at doses of 10 and 20 μ M, which stabilize HSF2, but neither globally impact the levels of proteasome-sensitive proteins (revised Supplementary Fig.11C, E,G; PCNA levels are not affected by proteasome inhibition by MG132), nor majorly alter iNPC or hCO differentiation (revised Supplementary Fig.11A upper left panels and D; staining by the neuroprogenitor cell markers, SOX2 and/or NESTIN for iNPCs and by SOX2 and the neuronal marker HuC/D for hCOs).

Both MG132-treated RSTS^{EP300} iNPCs and RSTS^{EP300} / RSTS^{CBP} hCOs exhibited increase in HSF2 levels, together with augmented levels of its targets, HSP110 and N-cadherin (revised Fig. 8A,B and Supplementary Fig. 11B). Since HSF2 is a major regulator of N-cadherin levels (Joutsen et al., 2020), this suggests that the CBP/EP300-HSF2-N-cadherin cascade is conserved in neurodevelopmental contexts. This CBP/EP300-to-HSF2 cascade also includes other HSF2 major targets like HSP110, an HSP involved in neurodevelopment, and protein quality control and stress chaperones, like HSP70.

Importantly, we performed two kinds of control experiments to show that the effect of MG132 is not due to a direct stabilization of the N-cadherin and HSP110 proteins, but acts through increase in HSF2 levels:

- First, we examine NDE1, an important neurodevelopmental player (Feng & Walsh, 2000, 2004), whose gene is a target of HSF2 (El Fatimy et al., 2014), but whose expression is not significantly regulated by HSF2 in the mouse developing cortex, (AdT unpublished data). Thus, this means that the levels of NDE1 protein should not increase upon MG132 treatment. As expected, we find that NDE1 protein levels do not exhibit major changes in response to MG132 exposure in the RSTS neural context (revised Fig. 8B and Supplementary Fig. 11B). Therefore, at these doses, MG132 does not globally stabilize every HSF2 gene target. This, combined to the fact that HSP110 and N-cadherin gene expression levels are strongly dependent on HSF2, suggests that, the elevation in HSP110 and N-cadherin levels upon MG132 treatment in the RSTS context is likely due to their strong positive regulator HSF2 and not by direct stabilization of these proteins by MG132 *per se*.
- Second, and importantly, MG132 treatment of HD hCOs neither significantly increases HSF2, HSP110, nor N-cadherin levels. Once again, this strongly supports the idea that the augmentation of N-cadherin and HSP110 in RSTS hCOs is not only due to the stabilization of these proteins by MG132 *per se*, but to the stabilization of their strong positive regulator HSF2 (Supplementary Fig. 11F,G).

Therefore, since HSF2 is a major regulator of N-cadherin levels (Joutsen et al., 2020), our data overall suggest that the CBP/EP300-HSF2-N-cadherin cascade that we identified in RSTS primary cells (hPSFs, Figure 7B-D), is active in neurodevelopmental contexts. Please see lanes 672-677 in the discussion.

*** Note that for HSP110, the unavailability of antibodies suitable for IF prevented us to perform IF.

4) Finally, we examined the potential functional impact of the CBP/EP300-to-HSF2 cascade on RSTS iNPC and hCO phenotypes. We have identified alteration of neurodevelopmental characteristics in RSTS-derived iNPCs and hCOs, compared to HD counterparts that are coherent with both mouse cortical *Hsf2*^{-/-} phenotypes (Chang et al., 2006) and cell-cell adhesion deficits. First, we observed perturbations in the radial organization of RSTS_{EP300} iNPCs, compared to HD iNPCs (Fig. 8C, which is reminiscent of the phenotype of mouse *Hsf2*^{-/-} cortices, which display perturbations of radial glia fiber organization (Chang et al., 2006). Second, we demonstrate that RSTS_{EP300} hCOs displayed a higher rate of ectopic mitoses, located at distance from the apical zone of the loops, compared to HD hCOs (Fig. 8D). A similar tendency is observed in RSTS_{CBP} hCOs. Importantly, N-cadherin and its partners at cell-cell adhesions are important regulators of proliferation and linked to the excessive production of mitotic progenitors located ectopically when impaired (reviewed by Hakanen et al., 2019; Liu et al., 2018). Please see pages 12 and 13, lanes 534-560 and pages 16 and 17, lanes 685-720.

Notably, most of the results were reproduced in 2 RSTS clones of different genetic origins (n=3 independent experiments), affecting either the *CREBBP* or the *EP300* gene, and corresponding to RSTS1 and RSTS2, respectively. In addition, concerning healthy donors, we used three completely distinct HD-derived cell sources: two HDs for hPSF studies and a different one for 2D and 3D neurodevelopmental models.

[REDACTED]

Finally, in support of our findings, the recent reporting of a deleterious *de novo* mutation of the *HSF2* gene linked to Angelman Syndrome (Aguilera et al., 2021) is a further proof of the importance of the integrity of the HSF2 pathway in neurodevelopment contexts.

REVIEWER #3

The authors presented data showing altered acetylation of HSF2 possibly instigates the neurodevelopmental deficits in Rubinstein-Taybi Syndrome (RSTS). They found that mutations of CBP or EP300 impair the stability of the short-lived HSF2 protein and in turn decreases the expression of genes that affect neural development. They also employed BTZ to upregulate HSF2 levels and restore the expression of HSP110 and N-Cadherin for developing treatment of RSTS. Their work reveals

additional regulatory machinery of neural development and its pathological relevance to RSTS. The strength of this work lies on their elaborative demonstration on the regulatory relationship between CBP and HSF. However, it also derails from further clarification of how the dynamic regulatory pathway causes the variable phenotypes of RSTS.

Several aspects need further clarification.

Point I. As HSF2 pleiotropically regulates neural development, such as controlling number of cortical radial glial cells and radial neuronal migration (lines 6-8 on page 5. It would be valuable if the authors present some data on this. Are there any relevant alterations in RSTS patients, animal models or brain organoids? Such kind of assays would be helpful to elucidate the machinery of RSTS and verify the effectiveness of the proposed treatment development.

1) HSF2, radial glia cells and RSTS. We thank the Reviewer for advising us to compare the mouse *Hsf2*^{-/-} phenotype of the developing neocortex to that of RSTS neural models. As mentioned by the Reviewer, we have reported i) decreased number of radial glia soma in the developing cortex of *Hsf2*^{-/-} fetuses, which is suggestive of decreased number of progenitor cells and ii) altered radial organization of radial glia fibers, accompanied by radial neuronal migration defects of the most superficial neuronal cortical layers (Chang et al., 2006). Here, we have chosen to work on human cerebral organoids (hCOs) rather than in mice for reasons linked to the Covid-19 sanitary situation (please see our answer to Reviewer #2, point II). We focused on early stages of hCOs and on potential proliferative niche phenotypes. We decided not to focus on radial neuronal migration, since hCOs are not appropriate to study the radial neuronal migration of neurons of the most superficial layers II and III (Velasco et al., 2020) which are specifically controlled by HSF2 (Chang et al., 2006).

We have thus put most energy in showing that disturbances of the CBP/EP300-HSF2-N-cadherin cascade is recapitulated in iNPCs and hCOs, in line with abnormal phenotypic traits linked to early stages of hCO differentiation. Indeed, we observed signs of disorganization of the proliferative niche, in line with perturbed radial organization of RSTS iNPCs stained by Pax6 and FABP7 in rosettes, compared to HD iNPCs (Fig. 8C). Likely, because physiologically decreased of N-cadherin expression leads to the onset of neuronal differentiation (Rousso et al., 2012) and that downregulation of N-cadherin levels, leads to imbalance self-renewal and premature differentiation (Iefremova et al., 2017), the phenotypic traits of RSTS hCOs may fairly reflect the impact of HSF2-dependent N-cadherin downregulation.

In keeping with parallels between the mouse adult *Hsf2*^{-/-} brain phenotypes and RSTS-derived cell models, we also have new data from *Hsf2*^{-/-}, showing that HSF2 regulates dendritic spine density in diverse brain regions, which is coherent with the work by Alari et al. (2018), showing reduced branch length and increased branch number in iNeuron 2D-cultures derived from RSTS patients (see below), as synaptogenesis and neuritogenesis share many mechanistic aspects relying on cytoskeleton organization.

[REDACTED]

[REDACTED]

2) Pharmacological manipulation of HSF2 levels in models of RSTS (see also Point II below). Normalizing the HSF2 pathway is an attractive target of therapeutic interest and, in particular for RSTS. BTZ would have been an interesting candidate for RSTS. Our revision work suggests that targeting HSF2 by BTZ is likely not the best way to achieve this goal, because BTZ, which has very powerful effects on HSF2 levels, even at very low (nM) doses in diverse cell systems (Rossi et al., 2014; Joutsen et al., 2020), including RSTS primary cells (revised Fig. 7D), did not increase HSF2 levels, compared to MG132, in our neural 2D and 3D cell models (please see our answer to Reviewer #2, Point II.3.b). This might be explained by the fact that tight regulation of HSF2 levels is important for mouse neurodevelopment (reviewed in Duchateau et al., 2020). A recent paper linking a deleterious *de novo* mutation in the human *HSF2* gene to Angelman Syndrome additionally supports the importance of intact HSF2 levels (Aguilera et al., 2021) in human neurodevelopment. As a consequence, compensatory mechanisms might counteract HSF2 upregulation by BTZ, in these neural 2D and 3D systems. In addition, BTZ and MG132 can differently operate through their mechanisms of inhibition of proteasomal activity and the reversibility of their action (Harbouri et al., 2017). Please, see page 12, lanes 504-532. We added a paragraph in the discussion, (page 16, lanes 677-684). Here, we thus have used pharmacology, more as a tool to validate the CBP/EP300-to-HSF2 cascade, than in an immediate therapeutic perspective. But we believe that targeting HSF2 by drugs with more controlled effects would be a valuable therapeutic approach, for which we are currently developing new approaches.

We therefore have mitigated our comments on BTZ as a therapeutic molecule for RSTS in page 15, lanes 644-645 of the manuscript and removed any comment about this point in the Discussion.

Point II. The authors show in RSTS cells that BTZ can increase the amount of N-Cadherin. They then propose such a drug can be further developed as a potential therapeutic drug. Additional assays, such as on mouse model of CBP/EP300 mutation or organoids of RSTS, plus in depth analysis are needed. It would be important to see in mouse model whether this drug can improve the cognitive function?

Tight regulation of HSF2 levels is important for mouse and human neurodevelopment (reviewed in Duchateau et al., 2020 Aguilera et al., 2021). So, modulating HSF2 levels should be performed in tightly controlled manner.

Thus, we used MG132 to allow for a limited increase in HSF2 levels, that is known to induce HSF2 levels, in a more limited manner, compared to BTZ in other cell systems (Rossi et al., 2014). We showed that treatment of RSTS iNPCs and hCOs by MG132 leads to elevated HSF2, HSP110, and N-cadherin levels (revised Fig. 8A,B and Supplementary Fig. 11A,B). Importantly, we made two kinds of control experiments to show that the effect of MG132 is not due to a direct stabilization of the N-cadherin and HSP110 proteins, but acts through increase in HSF2 levels (please see our answer to Reviewer #2, point II,3b):

- First, we examine NDE1, an important neurodevelopmental player (Feng & Walsh, 2000, 2004), which gene is a target of HSF2 (El Fatimy et al., 2014), but, whose expression, in contrast to N-cadherin and HSPs, is not significantly regulated by HSF2 in the mouse developing cortex, (AdT unpublished data). Importantly, NDE1 protein levels do not exhibit major changes in response to MG132 exposure in the RSTS neural context (revised Figure 8B and Supplementary Fig. 11B,E). This suggests that, at these doses, MG132 does not globally stabilize every HSF2 target and that, consequently, the elevation in HSP110 and N-cadherin levels upon MG132 treatment in the RSTS context likely occurs through increase in the levels of their strong positive regulator HSF2, and not by direct stabilization of these proteins by MG132 *per se*.

- Second, MG132 treatment of HD hCOs neither significantly increases HSF2, HSP110, nor N-cadherin levels, which, once again, supports the idea that the augmentation of N-cadherin and HSP110 in RSTS hCOs is not only due to the stabilization of these proteins by MG132 *per se*, but to the stabilization of their strong positive regulator HSF2 (Supplementary Fig. 11F,G).

We therefore think, since HSF2 is a major regulator of N-cadherin levels (Joutsen et al., 2020), that we have demonstrated that the CBP/EP300-HSF2-N-cadherin cascade is also active in these models of neurodevelopment.

2) The Reviewer's suggestion of using MG132 *in vivo* in mice and analyzing its impact in amelioration of mice cognitive function is of course highly relevant and attractive. However, it requires long preliminary experiments to determine appropriate doses and time-windows of drug exposure, followed by careful assessment of its effects on the developing brain before, eventually, explore cognitive functions by a battery of behavioral tests. In human, in RSTS patients, mutations in CREBBP or EP300 exist at the heterozygous state but, likely, exert dominant-negative effect on the intact remaining CBP

and EP300, which, to our knowledge has not yet been modelled in mice, yet. Indeed, the *Cbp*^{+/-} mouse model does not perfectly recapitulate the RSTS situation. In particular, we found no decrease in HSF2 activity in *Cbp*^{+/-} E15 cortices (AdT unpublished data; collaboration with the Angel Barco Laboratory). So, for our purpose, the best mouse model to mimic RSTS would be *Cbp*^{-/-} mice, but, *Cbp*^{-/-} fetuses are not viable (Yao et al., 1998; reviewed by Lipinski et al., 2019),

Nevertheless, the phenotypic traits of RSTS hCOs, which we have unveiled are not only coherent with the defects in that we observe cell-cell adhesion, but represent signatures of early neurodevelopmental defects, which are observed in other neurodevelopmental pathologies and are associated with impaired cognitive function, later in life (Courchesne et al., 2007). These include microcephaly (Lancaster et al., 2013), autism of different genetic origins which in defined time-windows of early cortical development, show accelerating production of cortical neurons (Paulsen et al., 2020). Although this would deserve a whole study on its own, our data suggest that the defects that we observe in early RSTS hCOs, especially those showing premature neuronal differentiation, could be compatible with the emergence of associated cognitive dysfunction in the mature brain. In addition, one de novo mutation in the *HSF2* gene has been recently associated with Angelman syndrome, which is characterized by severe developmental delay/ intellectual disability (Aguilera et al., 2021). Finally, three aspects are in agreement with behavioral impact of the lack of HSF2: i) the fact that HSF2 controls many genes involved in the formation and activity of synapses, as well as neuronal plasticity (our unpublished HSF2 ChIP-Seq data, Duchateau et al., in preparation); ii) the reduction dendritic spine density in various regions of *Hsf2*^{-/-} mice; see the enclosed figure above); iii) the role of HSF2 in NMDA-dependent neuroplasticity in the hippocampus (Driss et al., 2021).

As stated in our answer to Review#2 (Point II.4), we performed our study using hPSFs or iPSC clones derived from RSTS patients of two distinct genetic origins, one affecting the *CREBBP* gene and corresponding to RSTS1, the other affecting the *EP300* gene and corresponding to RSTS2. We also used three completely distinct HD-derived cell sources: two HDs for hPSF studies and a different one for 2D and 3D neurodevelopmental models. In addition, we robustly biophysically, biochemically and functionally highlighted and unraveled the CBP/EP300-HSF2-N-cadherin cascade in different systems, including primary skin fibroblasts from RSTS patients. Moreover, we now have recapitulated the cascade in iNPCs and hCOs from two RSTS patients of different genetic origin (mutated either in *CBP* or in *EP300*), and show that these RSTS hCOs exhibit phenotypic traits highly relevant for disturbances in N-cadherin expression. We thus think that our study overall supports a role for the CBP/EP300-to-HSF2 cascade and its deregulation in contexts modeling neurodevelopment, which opens new opportunities to understand and, in the long-term, provide therapeutic solutions for the RSTS pathology.

Point III. Some details need further clarity. The authors mentioned that HSF2 is acetylated by CBP/EP300 at three main lysine residues (K128, K135 and K197). In Fig. 2 and Fig. S2, only data of synthetic HSF2 peptide containing K82, K135 and K197 residues are shown, and no data about K128 residue is presented.

Indeed, this is a very good point. Actually, it turned out that the peptide containing the lysine K128 is insoluble and the *Proteogenix* Company repeatedly failed to synthesize this peptide. We had already included a comment on that point in the legend of revised Fig. 2F in the first submitted version of the manuscript:

“Note that it was not possible to investigate the acetylation of the HSF2 K128 peptide by CBP, because this peptide was repeatedly insoluble at the synthesis steps (Manufacturer’s information);
But, in agreement with the Reviewer’s suggestion, we now make this point clearer and more visible in the text. So, we have now transferred it to the main text of the “Results” section (page 7, lanes 282-283).

Point IV. In Fig.1 and Fig S1, although they show the general staining sites in organoid slices, there is no direct proof of the colocalization of CBP /HSF2 and markers of neural stem cells or neuronal cells. We have now performed confocal analyzes and included markers of NPCs and neurons both in hCOs and in the mouse cortex (revised Fig. 1A and revised Supplementary Fig. 1A). Please, see our answer to Reviewer #2, Point I.2).

Point V. The last three lines of P7, “impact of in silico mutations of the K177, K180, F181, V183 residues”, K180 should be Q180.

We thank the Reviewer and have corrected this in the text (page 9 of the revised manuscript) and also in revised Fig. 4D.

REVIEWER 4

HATs and HDACs conduct protein acetylation, including both histones and other proteins. It is believed that HATs and HDACs are important in brain development and also in neuron functionality. HDACs inhibitors have been long used in clinics to treat various neurological deficits including memory formation, mood, drug addiction, and neuroprotection, suggesting a role for HAT/HDAC in neuron homeostasis. In addition, mutations of the HAT complexes have been linked to human neurological deficits as well as to neurodevelopmental disorders. Rubinsten-Taybi Syndrome (RSTS) has been shown to be associated with mutations in HAT CBP/P300. However, how the epigenetic modulation, i.e., by CBP/P300, regulates brain development, remains largely unknown.

The authors devised molecular and biochemical experiments to decipher the molecular mechanism by which CBP/P300 regulates its downstream pathways. Intriguingly, the authors discovered that CBP/P300 directly interact and acetylate HSF2 (heat shock transcription factor), thereby its stability in response to heat shock or stress. The authors conducted comprehensive experiments, ranging from human brain organoid, mapping of protein-protein interactions and acetylation sites. They also used human (control donor and RSTS patients) materials and clinically approved drugs to verify the mechanisms that they discovered. The conclusion is supported by the data. The study presents a significant advance of the concept. However, some points should be addressed.

Specific comments

1. In order to view the colocalization of HSF2, CBP/P300 merge of different channels or stainings are needed. **Fig 1B-C**. The neuronal markers (eg. Tuj1) and NPC markers (eg. SOX2) should be provided in the same figure panels. **Fig S1B-C**. Merge images are needed.

As requested by the reviewer, and to better appreciate the co-localization between HSF2 and EP300/CBP, we have performed confocal analyzes of human cerebral organoids (hCOs) and added the HSF2/CBP_EP300 merge panels (revised Fig. 1A; please, see also our answer to Reviewer #2, Point I.2). In Supplementary figures, we kept separate (non-merged) panels as we think it also allows better visualization of the distribution of each protein, individually. In the same figure, we have also added a CBP/SOX2 (neuroprogenitor cells), a β III-TUBULIN (Tuj1; neurons)/PAX6 (neuroprogenitor cells) / CBP_EP300, as well as a CBP_EP300/TBR1 (young Neurons) merge panels.

In Supplementary Fig. 1A, we provide additional information about the colocalization of HSF2 and CBP/EP300 proteins in mouse embryonic cortex at E11 and E15, into which we have already characterized the expression profile of HSF2, both in neural progenitor cells and young neurons (Chang et al., 2006; El Fatimy et al., 2014; reviewed in Duchateau et al., 2020) and into which CBP/EP300 profiles have also been reported (Wang et al., 2010). These data further comfort the results that we have obtained in human brain organoids tissue, which show similar expression and co-localization profiles in human neurodevelopmental models, hCOs, and the mouse cortex. Please, see page 7, lanes 238-260.

2. **Fig 1E-F**. Many IP experiments showed separate Input blots. However, it is difficult to judge how good or abundant the interaction is. How many repeat experiments were done for Fig 1F?

We thank the reviewer for this comment. Please see also our answer to Reviewer #2, point 3.

Notably, co-immunoprecipitation experiments are only semi-quantitative and indicate that interaction is taking place in the cellular model or tissue of interest, but it is difficult to conclude about the strength of the interaction or the amount of protein involved into it. This is why we have deeply characterized the HSF2-CBP/EP300 by *in silico*, biophysical and biochemical approaches, as described in the following parts of the manuscript. Nevertheless, following the Reviewer's advice and to gain a better appreciation of interaction between HSF2 and CBP/EP300, we have reproduced the immunoprecipitation experiments in hCOs and run the inputs on the same gels (revised Figure 1E, the experiments were repeated 3 times; see also revised legends of figures). For ethical reasons, we did not reproduce all experiments carried out in mice, except for revised Fig. 1C. New and former experiments being considered, we have reproduced both co-IP experiments between HSF2 and CBP/EP300 in $n = 3$ independent experiments for HSF2 acetylation at two different stages at E10 and E15 in mice (revised Fig. 1D and Supplementary Fig. 1D, respectively). Importantly, as an additional indication of interaction between HSF2 and CBP, we have also observed interaction between CBP/EP300 and HSF2 in the human neuroblastoma line (SHSY-5Y) and added the inputs on the same panel with

immunoprecipitations samples (revised Supplementary Fig. 1E). Altogether, these results add support to the characterized tight interaction between HSF2 and CBP/EP300, which we extensively characterize in the manuscript, in neural contexts.

3. **Fig 3D-E.** How many cells scored for the percentages presented?

In the three independently performed experiments, the number of cells scored for the percentages presented in revised Fig. 3D,E for each condition ranged from 330 to 395 cells. We added this information in the corresponding legend in the revised manuscript. Note that to improve the readability of previous Figure 3D,E, we changed “Nb + cells (number of positive cells) into “% of positive cells”(revised Fig. 3).

4. **Fig 5E.** What are these levels (WT and 3KR) compared with 3KQ? P values need to be calculated and need more repeats (N=2 at the moment).

In revised Fig. 5D,E, using a pulse-chase experiment (see details in 5B), we report a ~50% decay in fluorescence intensity within 5 hours, in 2KO cells transfected with wild-type SNAP-HSF2 construct (SNAP-HSF2 WT), as a measure of the intrinsic decay of this protein. Preventing HSF2 acetylation (SNAP-HSF2 3KR) also resulted in a SNAP-HSF2 protein decay. Comparatively, mimicking HSF2 acetylation, using SNAP-HSF2 3KQ, protected the SNAP-HSF2 protein from decay.

In order to confirm that the SNAP-HSF2 WT and SNAP-HSF2 3KR protein decay was indeed due to proteasomal degradation of the protein, we treated these cells with the proteasome inhibitor MG132, and verified that this treatment blocked the SNAP-HSF2 WT and SNAP-HSF2 3KR decay, in a manner similar to what happened in untreated cells expressing the HSF2-acetylated mimic (SNAP-HSF2 3KQ). These experiments thus support the hypothesis that HSF2 is degraded by proteasome in this system, and stabilized by acetylation. We repeated the experiment and altogether we observed similar results in n=3 independent experiments (revised Figure 5E; new corresponding graph). We also explained the point of using MG132 in the legend of the revised Figure (please see page 26, lanes 1112-1115) and in the “Results” section (page 9, lanes 380-382).

5. **Fig 6.** It is clear that HSP70 is low with or without acute HS, compare to HD samples. However, after recovery, HSP70 was greatly increased and even with high magnitudes. It is also clear that HS did not induce high HSP70 in HD (CTR vs HS). I wonder how the authors quantified these signals (right panel). Is something wrong here? It was not correctly stated in the main text (page 9, para. 2).

We thank the reviewer for this relevant comment. It has long been well recognized that exposure to heat shock (HS) results in an increase in the HSP70 protein levels. The expression of this heat shock protein is typically analyzed 2h after the recovery phase, which is the time needed to reach its maximum (HS, Landry et al., 1982). This is the reason why we decided to put emphasis on its induction during the recovery period. However, as the reviewer noted, when the duration of HS is long (*i.e.* 1h, in our case), a slight increase of HSP70 can already be detected at the end of the HS. Consistently, only HD cells show a strong and significant induction of HSP70 during recovery but not RSTS cells.

- To be more representative of the quantified results, we now show another immunoblot for RSTS P1_{CBP}, revised Fig. 6D, left panels.

6. **Fig 7. (A)** The transfection efficiency and ectopic expression level of 3KQ should be controlled and documented. P values need to be calculated after total N number increased (currently N=2). **(D, F)** A quantification is needed, in order to make a firm statement.

1) As requested, we mentioned the transfection rate efficiency (*i.e.* 16.8% for HD and 10-12.4% for RSTS primary skin fibroblasts (hPSFs) in the legend of revised Fig. 7A (left panel).

2) In addition, we performed immunofluorescence experiments to document the ectopic expression of the HSF2 3KQ (revised Fig. 7A, right panel and Supplementary Fig. 7).

3) In order to quantify these experimental results, we performed a third experiment and calculated p-values (revised Fig. 7A). We also added a graph (Supplementary Fig. 8A), which corresponds to the quantification of the results, as a ratio of RSTS/HD, and thus allows a better appreciation of the restoration of nSB formation by HSF2 3KQ for each RSTS patient, compared to HD.

4) We quantified N-cadherin and HSP110 signals in the immunoblots presented in revised Fig. 7B,D (previous Fig. 7D,F), and, confirm the decrease of expression of both in RSTS compared to HD hPSFs as well as the rescue by BTZ on two protein levels.

7. **Fig S7.** Zoom in images would help to validate the statement.

As requested by the Reviewer, we now present a zoom for all images of Supplementary Fig. 8C (previously, Figure S7).

REFERENCES

- Courchesne E, Pierce K, Schumann CM, Redcay E, Buckwalter JA, Kennedy DP, Morgan J. Mapping early brain development in autism. *Neuron*. 2007 Oct 25;56(2):399-413. doi: 10.1016/j.neuron.2007.10.016. PMID: 17964254.
- Di K, Lloyd GK, Abraham V, MacLaren A, Burrows FJ, Desjardins A, Trikha M, Bota DA. Marizomib activity as a single agent in malignant gliomas: ability to cross the blood-brain barrier. *Neuro Oncol*. 2016 Jun;18(6):840-8. doi: 10.1093/neuonc/nov299. Epub 2015 Dec 17. PMID: 26681765; PMCID: PMC4864261.
- Drissi I, Deschamps C, Alary R, Robert A, Dubreuil V, Le Mouël A, Mohammed M, Sabéran-Djoneidi D, Mezger V, Naassila M, Pierrefiche O. Role of heat shock transcription factor 2 in the NMDA-dependent neuroplasticity induced by chronic ethanol intake in mouse hippocampus. *Addict Biol*. 2021 Mar;26(2):e12939. doi: 10.1111/adb.12939. Epub 2020 Jul 27. PMID: 32720424
- Hutton SR, Pevny LH. SOX2 expression levels distinguish between neural progenitor populations of the developing dorsal telencephalon. *Dev Biol*. 2011 Apr 1;352(1):40-7. doi: 10.1016/j.ydbio.2011.01.015. Epub 2011 Jan 21. PMID: 21256837.
- Lipinski M, Del Blanco B, Barco A. CBP/p300 in brain development and plasticity: disentangling the KAT's cradle. *Curr Opin Neurobiol*. 2019 Dec;59:1-8. doi: 10.1016/j.conb.2019.01.023. Epub 2019 Mar 8. PMID: 30856481.
- Miyamoto Y, Sakane F, Hashimoto K. N-cadherin-based adherens junction regulates the maintenance, proliferation, and differentiation of neural progenitor cells during development. *Cell Adh Migr*. 2015;9(3):183-92. doi: 10.1080/19336918.2015.1005466. Epub 2015 Apr 14. PMID: 25869655; PMCID: PMC4594476.
- Paulsen B, Velasco S, Kedaigle AJ, Pignoni M, Quadrato G, Deo A, Adiconis X, Uzquiano A, Kim K, Simmons SK, Tsafou K, Albanese A, Sartore R, Abbate C, Tucewicz A, Smith S, Chung K, Lage K, Regev A, Levin AZ, Arlotta P. Human brain organoids reveal accelerated development of cortical neuron classes as a shared feature of autism risk genes. *bioRxiv* 2020.11.10.376509; doi: <https://doi.org/10.1101/2020.11.10.376509>
- Rouso DL, Pearson CA, Gaber ZB, Miquelajauregui A, Li S, Portera-Cailliau C, Morrissey EE, Novitch BG. Foxp-mediated suppression of N-cadherin regulates neuroepithelial character and progenitor maintenance in the CNS. *Neuron*. 2012 Apr 26;74(2):314-30. doi: 10.1016/j.neuron.2012.02.024. PMID: 22542185
- Velasco S, Paulsen B, Arlotta P. 3D Brain Organoids: Studying Brain Development and Disease Outside the Embryo. *Annual Review of Neuroscience* 2020 43:1, 375-389. 10.1146/annurev-neuro-070918-050154
- Yao TP, Oh SP, Fuchs M, Zhou ND, Ch'ng LE, Newsome D, Bronson RT, Li E, Livingston DM, Eckner R. Gene dosage-dependent embryonic development and proliferation defects in mice lacking the transcriptional integrator p300. *Cell* 1998, 93:361-372.
- Zhang J, Woodhead GJ, Swaminathan SK, Noles SR, McQuinn ER, Pisarek AJ, Stocker AM, Mutch CA, Funatsu N, Chenn A. Cortical neural precursors inhibit their own differentiation via N-cadherin maintenance of beta-catenin signaling. *Dev Cell*. 2010 Mar 16;18(3):472-9. doi: 10.1016/j.devcel.2009.12.025. PMID: 20230753; PMCID: PMC2865854

REVIEWERS' COMMENTS

Reviewer #2 (Remarks to the Author):

The authors have addressed many key questions with extensive experiments. It is still unfortunate that the causality of HSF2 loss for downstream event could not be addressed clearly due to difficulty of conducting in utero gene transfer under the pandemic. Although many institutions in the US allow this experiment even under the pandemic this may be difficult in France. Many restrictions to lab experiments have been recently disappeared, and our normal scientific life has come back gradually. The clear proof of the causality (rescue experiments) will definitely improve the quality of the data. Via in utero gene transfer, the dosage of exogenous proteins can be well controlled (thus the problem that the authors faced might be escaped). The reviewer will trust further constructive discussion between the authors and editor in regard to this last missing portion.

Reviewer #3 (Remarks to the Author):

In the revised manuscript, all concerns raised by this reviewer have been addressed, either through additional data or by further explanation. The manuscript can be further processed for publication.

Reviewer #4 (Remarks to the Author):

The manuscript has been evaluated by 4 reviewers. The authors have taken serious steps to address all comments, particularly from three reviewers. The authors performed new experiments to consolidate their conclusions. My comments and concerns have been addressed in a satisfactory manner. The manuscript is greatly improved. I have no further questions.

RESPONSE TO REVIEWERS

REVIEWER 2 (REMARKS TO THE AUTHOR):

The authors have addressed many key questions with extensive experiments. It is still unfortunate that the causality of HSF2 loss for downstream event could not be addressed clearly due to difficulty of conducting in utero gene transfer under the pandemic. Although many institutions in the US allow this experiment even under the pandemic this may be difficult in France. Many restrictions to lab experiments have been recently disappeared, and our normal scientific life has come back gradually. The clear proof of the causality (rescue experiments) will definitely improve the quality of the data. Via in utero gene transfer, the dosage of exogenous proteins can be well controlled (thus the problem that the authors faced might be escaped). The reviewer will trust further constructive discussion between the authors and editor in regard to this last missing portion.

We thank the Reviewer for her/his comment and the Editor for his suggestion “*of including caveats, discussing limitation of conclusions that can be drawn in the absence of electroporation data* ». We have included a sentence to take it into account in the discussion (page 13, lanes 509-513):

“Finally, the various control experiments that we performed strongly suggest that the pharmacological rescue that we used is rather specific to HSF2 in terms of the restoration of N-cadherin and HSP110 expression. However, we are conscient that the clear proof of the causality of the defects in the CBP/EP300-HSF2-N-cadherin cascade on these phenotypes would require a stable and specific rescue of HSF2, based, for example, on genome-editing strategies.”

REVIEWER #3 (REMARKS TO THE AUTHOR):

In the revised manuscript, all concerns raised by this reviewer have been addressed, either through additional data or by further explanation. The manuscript can be further processed for publication.

We thank the Reviewer for her/his positive comments.

REVIEWER #4 (REMARKS TO THE AUTHOR):

The manuscript has been evaluated by 4 reviewers. The authors have taken serious steps to address all comments, particularly from three reviewers. The authors performed new experiments to consolidate their conclusions. My comments and concerns have been addressed in a satisfactory manner. The manuscript is greatly improved. I have no further questions.

We thank the Reviewer for her/his positive comments.